

# An inverse model to correct for the effects of post-depositional processing on ice-core nitrate and its isotopes: model framework and applications at Summit, Greenland and Dome C, Antarctica

Zhuang Jiang[1], Becky Alexander[2], Joel Savarino[3], Lei Geng[1,4,5*]

[1] Deep Space Exploration Laboratory/School of Earth and Space Sciences, University of Science and Technology of China, Hefei, Anhui, China

[2] Department of Atmospheric Sciences, University of Washington, Seattle WA, USA

[3] Univ. Grenoble Alpes, CNRS, IRD, G-INP, Institut des Géosciences de l'Environnement, Grenoble, France

[4] Laboratory for Ocean Dynamics and Climate, Pilot National Laboratory for Marine Science and Technology (Qingdao), Qingdao, Shandong, China

[5] CAS Center for Excellence in Comparative Planetology, University of Science and Technology of China, Hefei, Anhui, China

*Correspondence to*: Lei Geng (genglei@ustc.edu.cn)

## Abstract

Comprehensive evaluation of the effects of post-depositional processing is a prerequisite for appropriately interpreting ice-core records of nitrate concentration and isotopes. In this study, we developed an inverse model that uses archived snow/ice-core nitrate signals to reconstruct primary nitrate flux and its isotopes ($\delta^{15}$N and $\Delta^{17}$O). The model was then applied to two polar sites, Summit, Greenland and Dome C, Antarctica using measured snowpack nitrate concentration and isotope profiles in the top few meters. At Summit, the model successfully reproduced the observed atmospheric $\delta^{15}$N(NO$_3^-$) and $\Delta^{17}$O(NO$_3^-$) and their seasonality. The model was also able to reasonably reproduce the observed snowpack nitrate profiles at Dome C as well as the skin layer and atmospheric $\delta^{15}$N(NO$_3^-$) and $\Delta^{17}$O(NO$_3^-$) at the annual scale. The calculated F$_{pri}$ at Summit was $6.9 \times 10^{-6}$ kgN m$^2$ a$^{-1}$ and the calculated $\Delta^{17}$O(NO$_3^-$) of F$_{pri}$ is consistent with atmospheric observations in the northern hemisphere. However,





the calculated $\delta^{15}N(NO_3^-)$ of $F_{pri}$ displays an opposite seasonal pattern to atmospheric observations in the northern mid-latitudes, but is consistent with observations in two Arctic coastal sites. The calculated $F_{pri}$ at Dome C varies from 1.5 to 2.2 $\times 10^{-6}$ kgN m$^{-2}$ a$^{-1}$, with $\delta^{15}N(NO_3^-)$ of $F_{pri}$ varying from 6.2 to 29.3 ‰ and $\Delta^{17}O(NO_3^-)$ of $F_{pri}$ varying

from 48.8 to 52.6 ‰. The calculated $F_{pri}$ at Dome C is close to previous estimated stratospheric denitrification flux in Antarctica, and the high $\delta^{15}N(NO_3^-)$ and $\Delta^{17}O(NO_3^-)$ of $F_{pri}$ at Dome C also point towards the dominate role of stratospheric origin of primary nitrate to Dome C.

**1. Introduction**

Nitrate ion ($NO_3^-$) is routinely measured in polar snow and ice cores. The precursor of atmospheric nitrate is nitrogen oxides $NO_x$ (=$NO+NO_2$), which plays a fundamental role in the production of tropospheric ozone and interconversion of atmospheric $HO_x$ (=$OH+HO_2$) radicals (Seinfeld et al., 2016; Sillman, 1999). Given the potential link between ice-core nitrate and atmospheric $NO_x$, some previous studies proposed that ice-

core nitrate records could be used to derive information regarding past atmospheric $NO_x$ abundance (Dibb et al., 1998; Röthlisberger et al., 2000). In addition, the oxygen isotope mass-independent fractionation signal ($\Delta^{17}O = \delta^{17}O - 0.52 \times \delta^{18}O$) of nitrate is a reliable proxy of atmospheric $O_3/HO_x$ ratio and is directly related to atmospheric oxidizing environment (Alexander et al., 2004; Alexander et al., 2015; Geng et al., 2017; Sofen

et al., 2014). These unique features render ice core nitrate a potentially useful proxy to retrieve information on atmospheric oxidation environment in the past (Alexander et al., 2015).

Interpretations of ice-core nitrate records are, however, not straightforward (Wolff

et al., 2008). Unlike other less reactive species in ice cores such as sulfate or ammonia, ice-core nitrate may not be able to directly track its atmospheric abundance (Iizuka et al., 2018). To link ice-core nitrate to atmospheric $NO_x$ abundance, other information including the conversion rate of $NO_x$ to nitrate, the mean lifetime of atmospheric nitrate, and the impact of post-depositional processing must be considered (Wolff, 1995; Wolff



et al., 2008). Among these factors, the post-depositional processing of snow nitrate is the first gap in linking ice-core nitrate to atmospheric nitrate and/or $NO_x$.

Snow nitrate is reactive under exposure to sunlight and can be photolyzed to form $NO_x$ and HONO (Honrath et al., 2002; Chu and Anastasio, 2003), which is rapidly transported to the overlying atmosphere via diffusion and convection (Zatko et al.,
2013). These photoproducts subsequently reform nitrate (i.e., snow-sourced nitrate) and deposit locally or be exported away, leading to a recycling of nitrate at the air-snow interface (Erbland et al., 2013; Frey et al., 2009). This post-depositional processing not only disturbs the link between nitrate in snow and its atmospheric precursors but also alters its isotopic signals (Erbland et al., 2013; Jiang et al., 2021; Jiang et al., 2022; Shi
et al., 2015).

It is expected that the degree of post-depositional processing varies with changes in factors such as snow accumulation rate under different climates (Akers et al., 2022; Geng et al., 2015), causing corresponding shifts in the preserved nitrate signals. For example, the lower snow accumulation rate in glacial times would favor a higher degree
of post-depositional processing with elevated $\delta^{15}N(NO_3^-)$ relative to the Holocene as reflected by the GISP2 ice-core records (Geng et al., 2015; Hastings et al., 2005). Moreover, both observational and modeling studies have suggested that at sites with relatively high snow accumulation rates such as Summit, Greenland, the post-depositional processing of snow nitrate under present day conditions also has a
significant impact on seasonal $\delta^{15}N(NO_3^-)$ variations, although its integral effects at the annual scale are limited (Jiang et al., 2021; Jiang et al., 2022). In addition, the $\Delta^{17}O$ of snow nitrate would also be altered via secondary chemistry during photolysis on snow grain (i.e., the cage effect) and this effect is enhanced with lower snow accumulation rates (Erbland et al., 2013; Frey et al., 2009; Mccabe et al., 2005; Meusinger et al.,
2014). Thus, it is critical to evaluate the impact of post-depositional processing on ice core nitrate records before interpretation, especially for records covering different climates with changes in snow accumulation rates.

Primary nitrate to the polar ice sheets mainly originates from midlatitudes via long-



range transport and with extra contributions from stratospheric input (Lee et al., 2014;

Legrand and Delmas, 1986; Fischer et al., 1998; Savarino et al., 2007). To build the link

between ice-core and atmospheric nitrate, Geng et al. (2015) proposed a simple method

of using $\delta^{15}N(NO_3^-)$ to estimate the fractional loss of snow nitrate caused by post-

depositional processing. This method takes advantage of the high sensitivity of

$\delta^{15}N(NO_3^-)$ to the degree of photolytic loss (Erbland et al., 2013; Frey et al., 2009). If

$\delta^{15}N$ of the initially deposited nitrate can be assumed, the residual fraction of snow

nitrate can be calculated by applying a Rayleigh type isotope fractionation model. The

photolysis fractionation constant ($^{15}\varepsilon_p$) can be estimated via the prescribed actinic flux

spectrum and the absorption cross section for different nitrate isotopologues (Berhanu

et al., 2014). Based on this method, Geng et al. (2015) estimated that as much as 45-

53% of snow nitrate was lost after deposition during the last glacial time in the GISP2

ice core record. However, it's difficult to justify the assumed $\delta^{15}N$ of deposited nitrate

under different climates, and the method cannot correct for post-depositional

modification of $\Delta^{17}O(NO_3^-)$.

Erbland et al. (2015) developed a 1-D snow photochemistry model (TRANSITS,

https://github.com/JZxxhh/TRANSITS-model) that quantifies the effects of post-

depositional processing on the preservations of nitrate and its isotopes in ice cores. The

model comprises a series of physicochemical processes, including UV photolysis of

snow nitrate, emission of $NO_x$ to the overlying atmosphere, local oxidation and nitrate

deposition. In addition, changes in the isotopic composition of nitrate ($\delta^{15}N$ and $\Delta^{17}O$)

at each step of the post-depositional processing are also explicitly incorporated. The

model has been applied in various locations with different snow accumulation rates and

well reproduced the observed snowpack nitrate and isotope profiles (Erbland et al.,

2015; Jiang et al., 2021; Winton et al., 2020; Zatko et al., 2016). Based on model

sensitivity tests, Erbland et al. (2015) proposed a framework to correct for the effects

of post-depositional processing and to retrieve atmospheric information related to $F_{pri}$

at Dome C. However, the framework is rather complicated, and it assumes $\delta^{15}N$ of the

archived nitrate is exclusively determined by the degree of nitrate post-depositional




processing. Therefore, the framework cannot be applied to sites with moderate or high snow accumulation rates such as WAIS Divide, Antarctica and Summit, Greenland,

where factors other than post-depositional processing may also contribute to $\delta^{15}$N variations across different periods and/or climates (Hastings et al., 2005; Jiang et al., 2021).

In summary, TRANSITS is a forward model, and it requires prior knowledge of the distribution (e.g., weekly or monthly) of primary nitrate flux and isotopes as model

inputs, which is usually unavailable due to the lack of direct observations. In this study, we developed an inverse modeling framework (i.e., the inverse of the TRANISTS model) that uses snowpack and/or ice-core preserved nitrate signals (concentrations and isotopes) as model inputs, and properties of primary nitrate including its flux and isotopes ($\delta^{15}$N and $\Delta^{17}$O) can be directly retrieved with constraints from snow

accumulation rate and other known parameters (e.g., snow physicochemical properties). We assessed the model with observations at Summit, Greenland and Dome C, Antarctica, two representative sites with approximately the high-end and low-end snow accumulation rates at present day conditions.

## 2. Model description

**Table 1.** List of major parameters used in the inverse model.

| Compartment | Parameter | Unit | Definition |
|---|---|---|---|
| | FA | kgN m$^{-2}$ a$^{-1}$ | Archived nitrate flux |
| | $\delta^{15}$N(FA) | ‰ | $\delta^{15}$N of archived nitrate flux |
| | $\Delta^{17}$O(FA) | ‰ | $\Delta^{17}$O of archived nitrate flux |
| Input (could be | $A$ | kg m$^{-2}$ a$^{-1}$ | Snow accumulation rate |
| obtained from | $\rho$ | kg m$^{-3}$ | Snow density |
| measurements) | TCO | DU | Total column ozone |
| | LAI[a] | ng g$^{-1}$ | Light absorption impurities |
| | $\Phi$ | Dimensionless | Quantum yield of nitrate photolysis |
| | $\sigma$ | cm$^{-2}$ | Absorption cross section for NO$_3^-$ |
| Input | $\varepsilon_d$ | ‰ | Nitrogen isotope fractionation factor for nitrate deposition |
| (constrained by | $\Delta^{17}$O(FP) | ‰ | $\Delta^{17}$O of photolytic nitrate flux |
| observations) | $f_c$ | Dimensionless | Cage effect factor |
| | $f_{exp}$ | Dimensionless | Exported nitrate factor |
| Model output | F$_{pri}$ | kgN m$^{-2}$ a$^{-1}$ | Primary nitrate flux |





| | | |
|---|---|---|
| $\delta^{15}N(F_{pri})$ | ‰ | $\delta^{15}N$ of primary nitrate flux |
| $\Delta^{17}O(F_{pri})$ | ‰ | $\Delta^{17}O$ of primary nitrate flux |
| FD | kgN m$^{-2}$ a$^{-1}$ | Deposition nitrate flux |
| $\delta^{15}N(FD)$ | ‰ | $\delta^{15}N$ of deposition nitrate flux |
| $\Delta^{17}O(FD)$ | ‰ | $\Delta^{17}O$ of deposition nitrate flux |
| FP | kgN m$^{-2}$ a$^{-1}$ / ‰ | Photolytic nitrate flux |
| $\delta^{15}N(FP)$ | ‰ | $\delta^{15}N$ of photolytic nitrate flux |
| $\delta^{15}N(NO_3^-)_a$ | ‰ | $\delta^{15}N$ of local atmospheric nitrate |
| $\Delta^{17}O(NO_3^-)_a$ | ‰ | $\Delta^{17}O$ of local atmospheric nitrate |

$^a$Three types of light absorption impurity are considered in the inverse mode: black carbon, mineral dust and organic humic-like substance (HULIS).

The inverse model is designed based on the framework of the TRANSITS model but in an opposite direction of operating flows. The principle of the inverse model is that the archived snow nitrate concentration and isotope profiles from measurements are treated as model input, and they evolve inversely over time through the snow photic zone (defined as 3 times of the snow e-folding depth where the radiation decreases to 1/$e$ of its initial intensity at snow surface) to recover their initial states at the time of deposition , thus providing the initial isotope compositions and deposition fluxes before being affected by any post-depositional effects. The primary nitrate flux and its isotopes can be further obtained by solving the mass balance equations in the atmosphere box. A schematic view of the inverse model is shown in Fig. 1 with arrows pointing toward the model direction flow (i.e. inverse of the real physical processes). Major parameters in the inverse model and their definitions are listed in Table 1.



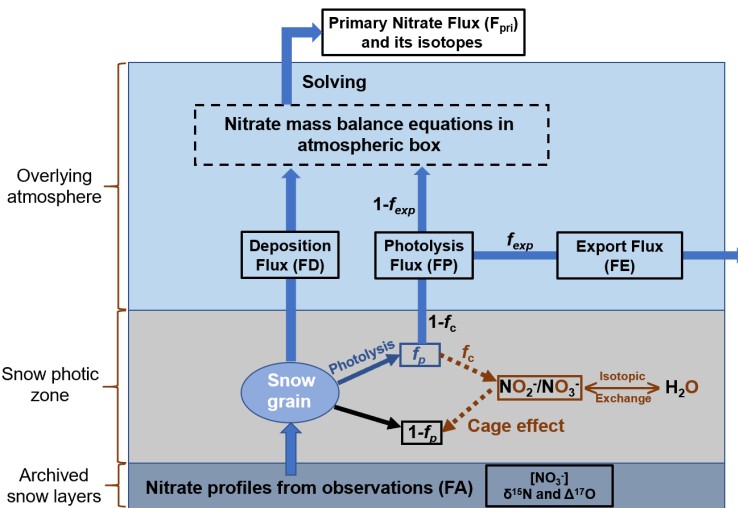

**Figure 1**. Schematic plot of the model domains of the inverse model including the atmospheric box, the snow photic zone and the archived snow layers, where $f_{exp}$ represents the fraction of nitrate exported from the site of photolysis. The nitrate isotopic and mass balance relationships on snow grains during photolysis are also shown, where $f_p$ represents the fraction of snow nitrate being photolyzed, and $f_c$ represents the fraction of photolyzed nitrate experiencing the cage effect (i.e., exchange of oxygen isotopes with snow water). FA represents the archived nitrate flux.

The inverse model inherits most of the original processes and features in TRANSITS but with several modifications. In accordance with the TRANSITS model, the domains of the inverse model are represented by a 1-D atmosphere and snow column. As shown in Fig. 1, the model contains three vertical parts, including the overlying atmospheric boundary layer which is treated as a single well-mixed box, and the underlying snowpack which is further separated into a snow photic zone and the archived snow layers beneath the photic zone. The model time step is set to be one week by default. During each time step, the mass conservation equations in the atmospheric box are represented as follows:

$$\frac{\mathrm{d}m_a}{\mathrm{d}t} = F_{pri} + FP - FE - FD \tag{1}$$

$$\frac{\mathrm{d}(m_a \times \delta^{15}N_a)}{\mathrm{d}t} = F_{pri} \times \delta^{15}N(F_{pri}) + FP \times \delta^{15}N(FP) - $$
$$FE \times \delta^{15}N(FP) - FD \times \delta^{15}N(FD) \tag{2}$$





$$\frac{\mathrm{d}(m_a \times \Delta^{17}O_a)}{\mathrm{dt}} = F_{pri} \times \Delta^{17}O(F_{pri}) + FP \times \Delta^{17}O(FP) - $$
$$FE \times \Delta^{17}O(FE) - FD \times \Delta^{17}O(FD) \qquad (3)$$

where the subscript "a" represents the atmospheric box, i.e., $m_a$ refers to the mass of

atmospheric nitrate, and $\delta^{15}N_a$ and $\Delta^{17}O_a$ refer to $\delta^{15}N$ and $\Delta^{17}O$ of atmospheric nitrate,

respectively. Different nitrate fluxes transported in and out of the atmospheric box are

denoted as FP, FE, and FD, where FP refers to the photolytic nitrate flux (the snow-

sourced nitrate), FD refers to the atmospheric deposition nitrate flux, and FE refers to

the exported nitrate flux that is horizontally transported out of the atmospheric box via

air flow. Following Erbland et al. (2015), FE is assumed to be a portion ($f_{exp}$) of FP (i.e.,

FE = $f_{exp}$ × FP) and maintains the isotopic signatures of FP.

In Eq. (1-3), the LHS (left-hand side) terms are two to three orders of magnitude

smaller than nitrate fluxes in and out of the atmospheric box. Erbland et al. (2015)

showed that the atmospheric nitrate mass was a factor of ~$10^{-3}$ smaller than the surface

snow nitrate reservoir at Dome C, and similar results were also found at Summit in

Jiang et al. (2021). Thus, d(x)/dt is assumed to be zero at each time step (i.e. species

and isotope compositions in the atmosphere are considered at steady state), which leads

to simplified formulas for calculating $F_{pri}$ via Eq. (4-6) as follows:

$$F_{pri} \approx FD - FP(1 - f_{exp}) \qquad (4)$$

$$\delta^{15}N(F_{pri}) \approx \frac{FD \times \delta^{15}N(FD) - FP \times (1 - f_{exp}) \times \delta^{15}N(FP)}{FD - FP \times (1 - f_{exp})} \qquad (5)$$

$$\Delta^{17}O(F_{pri}) \approx \frac{FD \times \Delta^{17}O(FD) - FP \times (1 - f_{exp}) \times \Delta^{17}O(FP)}{FD - FP \times (1 - f_{exp})} \qquad (6)$$

Hence, if the magnitude and isotopic compositions of FP and FD in each time step

are known, $F_{pri}$ can be calculated. FP and FD are calculated from the inverse evolution

of snowpack nitrate are described in the following sections.

### 2.1 The backward evolution of snowpack nitrate

Starting with an arbitrary snowpack nitrate depth profile at a given time step,

changes in nitrate concentration and isotopic compositions ($\delta^{15}N$ and $\Delta^{17}O$) in a certain

snow layer in the photic zone induced by photolysis can be calculated as follows:



$$c(SN_n') = \frac{c(SN_n)}{(1 - f_p) + f_c f_p} \quad (7)$$

$$\delta^{15}N(SN_n') = \delta^{15}N(SN_n) - \frac{(1 - f_p)(1 - f_c)\overline{\varepsilon_p}ln(1 - f_p)}{(1 - f_p) + f_c f_p} \quad (8)$$

$$\Delta^{17}O(SN_n') = \Delta^{17}O(SN_n) \frac{(1 - f_p) + f_c f_p}{(1 - f_p) + \frac{2}{3}f_c f_p} \quad (9)$$

where $c$ represent the nitrate concentration and $SN_n$ refers to the $n^{th}$ snowpack layer, respectively, and the quotation mark in superscript refers to the initial state before being photolyzed at each time step. These equations are based on the nitrate mass and isotopic balances on snow grains during photolysis as shown in Fig. 1, and detailed derivations of these equations can be found in Appendix A.

In Eq (7-9), $f_p$ represents the fraction of snow nitrate that undergoes photolysis at each time step, and $f_c$ represents the fraction of nitrate photolysis intermediate undergoing the cage effect (Meusinger et al., 2014) which leads to apparent oxygen isotope exchange with water and lowers $\Delta^{17}O$ by a factor of 2/3. The potential isotope effect on $\delta^{15}N$ during cage effect remains unknown and is not considered. The value of $f_p$ is calculated by the first-order reaction of nitrate photolysis:

$$f_p = 1 - \exp\left(-\int_0^{dt} J(t,z)dt\right) \quad (10)$$

where $J$ represents the rate constant of nitrate photolysis that varies with time and depth of the snow layer. $J$ is calculated from actinic flux ($I$), the quantum yield ($\Phi$), and the absorption cross section ($\sigma$) of nitrate photolysis as follows:

$$J(t,z) = \int_{280\,nm}^{350\,nm} \Phi(\lambda) \times \sigma_{NO_3^-}(\lambda) \times I(z,\lambda)\, d\lambda \quad (11)$$

The rate constant of $^{15}NO_3^-$ photolysis ($J^*$) is also calculated from the absorption cross section of the heavy isotopologue from Berhanu et al. (2014), and the photolysis fractionation constant for nitrogen isotope $\varepsilon_p$ is calculated via:

$$\varepsilon_p(t,z) = \frac{J^*(t,z)}{J(t,z)} - 1 \quad (12)$$

The solar zenith angle changes with time during each time step, leading to changes





in the spectrum of actinic flux and subsequently changes in $\varepsilon_p$. To simplify the calculation, in Eq. (8) the weighted average of nitrogen isotope fractionation constant ($\overline{\varepsilon_p}$) over different solar zenith angle is used. The radiative transfer in snowpack is

calculated using the parameterization from Zatko et al. (2013) to achieve fast online calculations, and this parameterization has been shown to be capable of providing consistent results with a high-order snowpack radiative transfer model DISORT (Zatko et al., 2013). The upper boundary conditions for the parameterization, i.e., the direct and diffuse components of the irradiance at the snow surface, are calculated offline

using the Troposphere Ultraviolet and Visible (TUV) radiation model (Madronich et al., 1998) at different total column ozone (TCO) and solar zenith angle conditions.

The relationships between $c(SN_n)$, $\delta^{15}N(SN_n)$, $\Delta^{17}O(SN_n)$ and $c(SN_n')$, $\delta^{15}N(SN_n')$, $\Delta^{17}O(SN_n')$ in the snowpack are illustrated in Fig. 2, where $c(SN_n)$, $\delta^{15}N(SN_n)$, $\Delta^{17}O(SN_n)$ are the values after photolysis in the $n^{th}$ layer at a certain time step, and $c(SN_n')$,

$\delta^{15}N(SN_n')$, $\Delta^{17}O(SN_n')$ are the values before photolysis (calculated by Eq (7-9)) and are also the values after photolysis in the prior time step when it was in the $(n-1)^{th}$ layer. By repeating this operation, the initially deposited values of nitrate concentration and isotopes for a given snow layer without influence from the photo-driven post-depositional processing (i.e., when this layer was at the surface) can be calculated,

which is be further linked to FD.

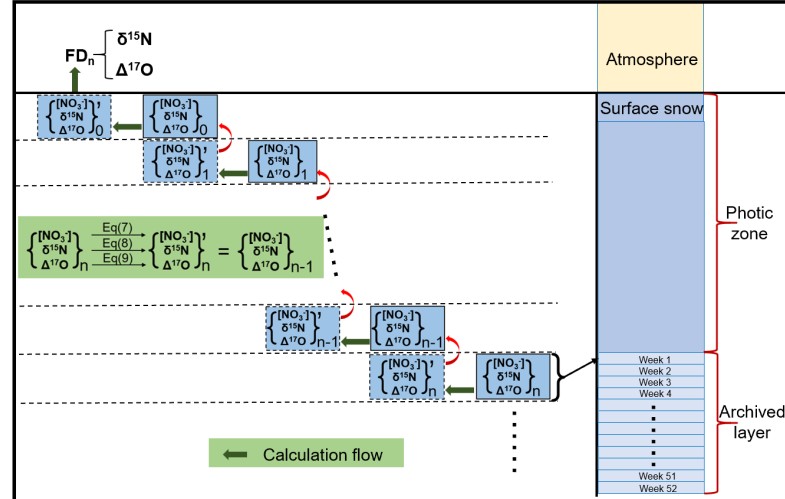





**Figure 2**. Schematic of the evolution of snowpack nitrate from the archived layer to its initial state at the snow surface. The quotation mark in the superscript of the bracket represents the status of snow nitrate before photolysis at each time step.

**2.2 Determinations of FP and FD**

FP and FD are determined during the inverse evolution of snowpack nitrate profiles. As shown in Fig. 1, photolytic nitrate flux and its $\delta^{15}N$ from the $n^{th}$ snow layer can be calculated via the mass balance relationships:

$$FP_n = c(SN'_n)f_p(1 - f_c)\rho_{snow}d_n/\Delta t \qquad (13)$$

$$\delta^{15}N(FP_n) = \delta^{15}N(SN_n) - \frac{\overline{\varepsilon_p}(1 - f_p)ln(1 - f_p)}{f_p} \qquad (14)$$

In Eq (13), $\rho_{snow}$ is the density of snow, $d_n$ is the thickness of the $n^{th}$ snow layer, which is equal to the accumulated snow thickness at one time step, and $\Delta t$ is the default model time step (1 week). Eq (14) implicitly assumes that the reformed nitrate in the overlying atmosphere keeps the same $\delta^{15}N$ signals of the snow-emitted photoproduct of $NO_2$ because of isotope mass balance, i.e., essentially all $NO_2$ is oxidized into nitrate at one time step. FP emitted from the whole snowpack and its $\delta^{15}N$ can be calculated by:

$$FP = \sum FP_n \qquad (15)$$

$$\delta^{15}N(FP) = \frac{\sum FP_n * \delta^{15}N(FP_n)}{\sum FP_n} \qquad (16)$$

For $\Delta^{17}O$ of FP, extra knowledge of the oxidizing agent concentrations in the local atmosphere including $HO_2$, $RO_2$ and $O_3$ must be provided (Appendix B). This is because the emitted $NO_x$ would achieve photochemical steady state rapidly, thus erasing any original $\Delta^{17}O$ signal inherited from the snowpack nitrate. The subsequent conversion of $NO_2$ to nitrate would also determine 1/3 of the oxygen atom of the newly formed nitrate.

FD and its isotopic signals can be obtained from the uppermost snow layer before photolysis occurs as illustrated in Fig.2:

$$FD = c(SN'_0)\rho_{snow}d_0/\Delta t \qquad (17)$$

$$\delta^{15}N(FD) = \delta^{15}N(SN'_0) \qquad (18)$$





$$\Delta^{17}O(FD) = \Delta^{17}O(SN_0')$$ (19)

The calculated FP and FD in each time step are further used to calculate $F_{pri}$ according to Eq (4-6).

### 2.3 The choice of model initial conditions

To run the model, an appropriate archival snow nitrate profile with known concentration and isotopic composition ($\delta^{15}N$ and $\Delta^{17}O$) should be assigned as model initial conditions with seasonal or monthly resolution, though ideally weekly or finer resolution data are the best. The archived nitrate profile could be dated by using various types of seasonal markers, such as the $\delta^{18}O$ of $H_2O$, the ion concentrations or their ratios, and the snow accumulation rates (Hastings et al., 2004; Furukawa et al., 2017; Dibb et al., 2007).

### 3. Model evaluations

Because there lack direct observations of primary nitrate, we evaluated the model performance with other kinds of observations, including nitrate isotopes in surface snow and the overlying atmosphere. The deposited nitrate flux FD represents the state of nitrate that has just deposited onto the surface snow and is close to the definition of the skin layer of snowpack, i.e., the uppermost several millimeters of surface snow (Erbland et al., 2013; Winton et al., 2020). Thus, if there are sufficient high-resolution skin layer observations, a direct comparison with the model output can be performed (i.e., FD vs. skin layer measurements). Moreover, since FD originates from the local atmosphere, if the air-snow nitrate transfer function (i.e., the mass and isotope relationships between atmospheric nitrate and the deposited nitrate) is known, the calculated FD could be used to infer the state of local atmospheric nitrate. In this study, the isotope transfer function is applied instead of the mass transfer function because of its simplicity, especially for $\Delta^{17}O$, which is assumed to be conserved during deposition owing to its mass-independent nature. For $\delta^{15}N$, we assume that the deposition of atmospheric nitrate is associated with a fractionation constant ($\varepsilon_d$) of +10 ‰ following Erbland et al. (2013). In all, we can either directly compare the modeled isotopes of FD with the observed values in the skin layer or with local atmospheric signals by including





the differences (only for $\delta^{15}$N) between FD and atmospheric nitrate.

In this study, we chose two typical polar sites, Summit, Greenland, and Dome C, Antarctica to conduct case studies in order to test the performance of the inverse model. These two sites were chosen for several reasons. First, these two sites represent typical polar sites with both relatively high (Summit) and extremely low (Dome C) snow accumulation rates. Second, there are sufficient atmospheric and/or snow observations

at these two sites, which informs model input parameters and allows for comparison of the model results with observations. Third, these two sites are hot spots of ice core drilling, and future work using the inverse model on ice core nitrate records from these sites can be performed. In addition, there have already been studies simulating the post-depositional processing of snow nitrate at these two sites by using the forward

TRANSITS model (Erbland et al., 2015; Jiang et al., 2021). Most of the model parameters in this study are kept the same as the original TRANISTS simulation unless otherwise mentioned. The major parameters used in this study are summarized in Table 2. Below, we specifically describe how we chose the initial model values/conditions for simulations at these two sites.


**Table 2.** Values of major parameter used in the model simulations at two different sites.

| Parameter | Dome C, Antarctica | | Summit, Greenland | |
|---|---|---|---|---|
| | Value | Reference | Value | Reference |
| $A$ | 28 kg m$^{-2}$ a$^{-1}$ | Erbland et al. (2015) | 250 kg m$^{-2}$ a$^{-1}$ | Dibb et al., (2014) |
| $\Phi$ | 0.015 | Adjusted[a] | 0.002 | Jiang et al., (2021) |
| $f_c$ | 0.15 | Erbland et al. (2015) | 0.15 | Erbland et al. (2015) |
| $f_{exp}$ | 0.2 | Erbland et al. (2015) | 0.35 | Jiang et al., (2021) |
| $\varepsilon_d$ | +10 ‰ | Erbland et al. (2013) | +10 ‰ | Erbland et al. (2013) |
| $\Delta^{17}O(NO_3^-)$ of FP | Calculated | Erbland et al. (2013) | Observed atmospheric $\Delta^{17}O(NO_3^-)$ | Jiang et al., (2021) |

[a]Adjusted according to the best fit of snowpack nitrate $\delta^{15}$N profile at Dome C (Appendix C).



### 3.1 Summit, Greenland

Summit, Greenland is a typical site with high snow accumulation rate (250 kg m$^{-2}$ a$^{-1}$, Dibb et al., 2004) at present, and weekly resolved snow accumulation data exists (Burkhart et al., 2004), allowing for the precise dating of the snowpack nitrate profile (Jiang et al., 2022). The weekly archival snowpack nitrate records at Summit compiled in Jiang et al. (2022) were adapted as initial model values.

### 3.2 Dome C, Antarctica

The present snow accumulation rate at Dome C, Antarctica is extremely low (28 kg m$^{-2}$ a$^{-1}$, Erbland et al., 2013), and it is currently impossible to discern seasonal or sub-seasonal nitrate patterns owing to the limited resolution of snowpack measurements. Erbland et al. (2013) reported five snowpack nitrate depth profiles at Dome C that extended just below the photic zone. To predict the final archived nitrate concentration

and isotopes, Erbland et al. (2013) fitted these depth profiles with an exponential function, and the obtained asymptotic values were regarded as the final preserved nitrate signal. The average asymptotic values for the five snowpacks were $(21.2 \pm 18.1)$ ng g$^{-1}$, $(273.6 \pm 64.0)$ ‰ and $(26.0 \pm 1.9)$ ‰ for nitrate concentration, $\delta^{15}$N and $\Delta^{17}$O,

respectively. These values were used as the annual averages of the preserved nitrate at Dome C in this study.

        We note the seasonality of the archived nitrate concentration is important because it determines the magnitude of FP and FD at each time step in the model. In simulations of Dome C, we designed three cases with different weekly concentration distributions

in a year. In case 1, the weekly nitrate concentrations were assumed to be uniform throughout a year. In case 2, the weekly archival nitrate concentrations were assumed to be a Gaussian-type distribution to match the observed seasonality in skin layer nitrate concentrations at Dome C (Erbland et al., 2013):

$$c(n) = c_a \times \left( a + b \times \exp\left( -\frac{(n - n_0)^2}{\sigma^2} \right) \right) \qquad (20)$$

In Eq (20), $c_a$ represents the annual average snow nitrate concentration, $n$ represents the week number (1 to 52) and the shape parameters (a, b, σ) were determined by the best fit of skin layer nitrate concentrations (Appendix D). $n_0$ represents the week when





nitrate concentration peaks in a year and was set to be 26 according to the observed
maximum nitrate concentrations in the skin layer in local midsummer (Erbland et al.,
2013). However, since nitrate deposited in different weeks of a year would have
experienced different amounts of total actinic flux and nitrate deposited in autumn
undergoes minimal degree of photolysis (Jiang et al., 2022), it is likely that the summer
peak would shift toward autumn by final preservation. As such, we also prescribed a
"shifted peak" distribution in case 3, and in this case $n_0$ was set equal to 35 in Eq. (20),
while other parameters were the same as in case 2.

To determine the uncertainties in the model results caused by these artificially
assumed nitrate profiles, we applied a Monte Carlo method, i.e., the exact initial value
in snow at each week was set arbitrarily as follows:

$$c_r = c_a + \mathbf{U}(-\sigma, \sigma) \tag{21}$$

where $c_a$ represents the prescribed initial value of annual-mean snow nitrate
concentration in each case as described above, $\mathbf{U}$ represents a uniformly distributed
random variable and $\sigma$ represents the standard error of the observed ca. The obtained
time series with random error was normalized again as final model inputs. All three
cases were repeated 1000 times, and the model results were used to evaluate the
uncertainties.

For isotopic ratios ($\delta^{15}$N and $\Delta^{17}$O) of the archived nitrate, their seasonality was
omitted in this study to simplify the model calculations, and for the results at Dome C
we only compared the modeled results with observations at annual scale given the
unknown seasonal inputs of these parameters. Note that reconstruction of atmospheric
signals of $\delta^{15}$N and $\Delta^{17}$O from ice core records usually use a coarser resolution than
sub-annual variations, justifying our annual averaging approach.

## 4. Results and discussion

### 4.1 Model results at Summit, Greenland



### 4.1.1 Comparison of local atmospheric variations

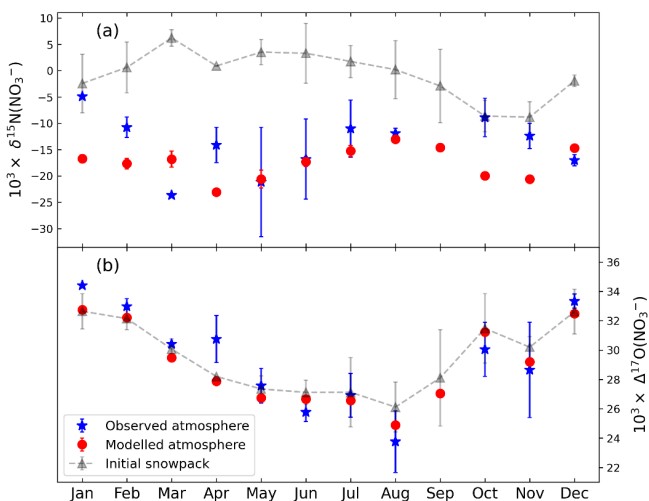


**Figure 3.** Comparison between the modeled (red dots) and observed (blue stars) seasonal variations in atmospheric nitrate (**a**) $\delta^{15}N$ and (**b**) $\Delta^{17}O$ at Summit Greenland. The dashed line with gray triangle represents the snowpack measurements as model inputs (i.e., the monthly archived snow nitrate $\delta^{15}N$ and $\Delta^{17}O$ from snowpack

observations) (Jiang et al. 2022). The atmospheric observations were from Jiang et al. (2022).

Currently there are no skin layer observations at Summit and the atmospheric nitrate isotopes reported by Jiang et al. (2022) were used for model comparison. To reduce the uncertainty of model results owing to uncertainties associated with the

weekly dating of snowpack, we only compared the modeled monthly averages with observations, and the uncertainties of the monthly model results were calculated as one standard error of the mean of results from different weeks. As shown in Fig. 3, the modeled seasonal variations in atmospheric $\delta^{15}N(NO_3^-)$ and $\Delta^{17}O(NO_3^-)$ agree well with the observed seasonality. In addition, the modeled and observed atmospheric

$\Delta^{17}O(NO_3^-)$ are both close to snowpack $\Delta^{17}O(NO_3^-)$. This is just as expected at Summit since the deposition of atmospheric nitrate is assumed to be a mass-dependent process. The only process that can alter $\Delta^{17}O(NO_3^-)$ in snow is the cage effect (McCabe et al., 2005), which is negligible under present Summit conditions. The inverse model calculated a small cage effect of 0.15 ‰ on $\Delta^{17}O(NO_3^-)$ by comparing the annual

weighted average of FA and FD, which is close to the value of 0.19 ‰ predicted by the





TRANSITS model (Jiang et al., 2021).

For $\delta^{15}N(NO_3^-)$, the modeled atmospheric $\delta^{15}N(NO_3^-)$ seasonality is comparable to the observations, but the absolute values display some discrepancies in autumn and winter. In particular, the modeled and observed average $\delta^{15}N(NO_3^-)$ values in the

summer half-year (from March to August) are -17.6 ± 3.5 ‰ and -16.0 ± 7.8 ‰, respectively, while in the winter half-year they are -16.0 ± 7.8 ‰ and -12.0 ± 4.1 ‰, respectively. The model-observation difference in the winter half year may be related to the model set-up of a constant $\varepsilon_d$ of +10 ‰. As discussed by Jiang et al. (2022), the partition between nitrate deposition mechanisms (i.e., wet vs. dry deposition) may result

in seasonally different air-snow transfer functions for $\delta^{15}N(NO_3^-)$. It has been observed that $\delta^{15}N(NO_3^-)$ of dry deposition is generally higher than wet deposition (Beyn et al., 2014; Heaton, 1987). This implies that dry deposition likely possesses a larger $\varepsilon_d$, perhaps because wet deposition can scavenge all or most of atmospheric nitrate, leading to small to no isotope fractionation. Given the potential seasonal changes in the relative

fraction of dry versus wet deposition at Summit, using a constant $\varepsilon_d$ in the model would likely cause discrepancies in one season but not in the other. Some observations at Summit indicate that snowfall activities are more frequent and severe in summer months (June-September) than in winter (Castellani et al., 2015; Bennartz et al., 2019), which implies that dry deposition of atmospheric nitrate is more important in winter

instead of summer. Thus, the model-observation discrepancies in the winter half year cannot be explained by the seasonal shift in the ratio between wet and dry deposition, as more dry deposition should result in a larger isotope effect in winter.

Alternatively, we note the $\varepsilon_d$ itself may have a seasonality which could be caused by the temperature dependence of nitrate absorption onto ice grains (Abbatt, 1997) or

be influenced by other mechanisms such as the stability of the boundary layer. In fact, observations at Dome C indicated the averaged enrichment in skin layer $\delta^{15}N(NO_3^-)$ related to atmospheric $\delta^{15}N(NO_3^-)$ is +25 ‰ in summer, while in winter the value is +10 ‰ (Erbland et al., 2013). This may indicate a larger $\varepsilon_d$ in summer than in winter, though the summer skin layer $\delta^{15}N(NO_3^-)$ is probably more or less influenced by




photolysis which tends to increase $\delta^{15}N(NO_3^-)$. However, in the inverse model, $\varepsilon_d$ was

set as +10 ‰ throughout the year for Summit (note this value is consistent with the

observed difference between surface snow and local atmospheric $\delta^{15}N(NO_3^-)$ at Summit

in May and June (Fibiger et al., 2016)). If at Summit the $\varepsilon_d$ in winter is lower than that

in summer, the modeled average $\delta^{15}N(NO_3^-)$ in the winter half year would have been

underestimated. This at least explains in part the model-observation discrepancies in

winter half year $\delta^{15}N(NO_3^-)$. Future work on the degree of nitrogen isotope fractionation

during atmospheric nitrate deposition and the causal factors are necessary to further

investigate this issue.

### 4.1.2 Flux and isotopes of primary nitrate

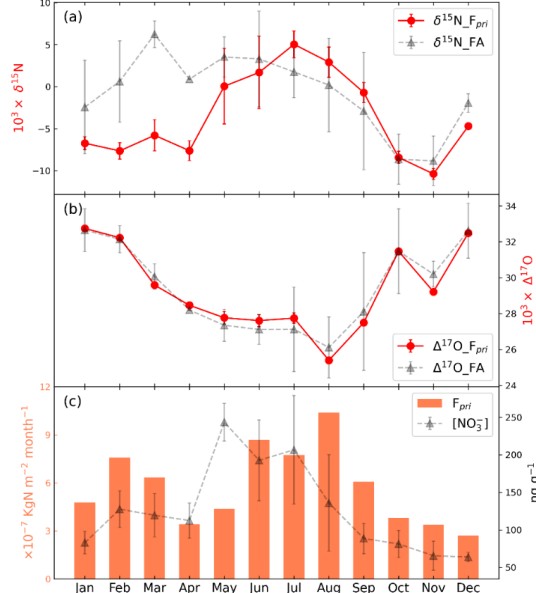


**Figure 4.** The modeled monthly (**a**) $\delta^{15}N$, (**b**) $\Delta^{17}O$ and (**c**) fluxes of primary nitrate
($F_{pri}$) to Summit Greenland. The concentration, $\delta^{15}N$ and $\Delta^{17}O$ values of the archived
snow nitrate as model inputs are also shown for comparison (gray triangle with
dashed line).


The major function of the inverse model is to reconstruct the primary nitrate flux

and its isotopes by using ice core nitrate records. Primary nitrate flux is closely

associated with atmospheric nitrate at a regional scale and could be further linked to the



atmospheric abundance of its precursor $NO_x$. The isotopic composition of $F_{pri}$ could provide extra information. For example, the $\delta^{15}N$ of $F_{pri}$ may be used to infer the variations in $NO_x$ source emissions if other factors influencing isotope fractionation during the atmospheric conversion of $NO_x$ to nitrate can be constrained. The $\Delta^{17}O$ of $F_{pri}$ depends on the relative concentration of major atmospheric oxidations such as $O_3$ and $HO_2/RO_2$ radicals and thus could be used to reflect regional atmospheric oxidation environment (Geng et al., 2017; Sofen et al., 2014).

The model-calculated seasonal variations in $F_{pri}$ to Summit are shown in Fig. 4. The annual flux of primary nitrate was calculated to be $6.96 \times 10^{-6}$ kgN $m^2$ $a^{-1}$, which is similar ($\approx$ 2-3 $\times 10^{-6}$ kgN $m^2$ $a^{-1}$) to model results from Zatko et al. (2016) using the GEOS-Chem model but is about two orders of magnitude lower than the atmospheric nitrate deposition flux in mid-latitude area (Gao et al., 2019; Zhang et al., 2012). The seasonality of $F_{pri}$ displays a bimodal mode with a major summer peak and a secondary peak in late winter/early spring, in contrast the preserved snowpack nitrate concentration which peaks in spring/summer. The maximum $F_{pri}$ in summer could be caused by the enhanced temperature-dependent precursor $NO_x$ emissions such as from soil microbes (Pilegaard et al., 2006) as well as the more active photochemistry in summer, both of which would promote more efficient atmospheric nitrate production. It is interesting that the secondary $F_{pri}$ peak in early spring is coincident with the timing of the spring Arctic haze phenomenon (Quinn et al., 2007), as well as the occasional spring nitrate concentration peak in snowpack and ice cores at Summit (Geng et al., 2014), though the exact timing of the seasonal peaks needs further investigation.

The modeled $\Delta^{17}O$ of $F_{pri}$ is close to the measurement in snowpack with minimum values in summer, suggesting the $\Delta^{17}O$ signal of primary nitrate is well preserved under current Summit conditions. The seasonal variations in $\Delta^{17}O$ of $F_{pri}$ can be understood in terms of the different production mechanisms of atmospheric nitrate (Alexander et al., 2020). In summer, ample solar radiation enhances the photochemical production of $HNO_3$ from the $NO_2+OH$ pathway, the $\Delta^{17}O$ of which is lowest compared with other nitrate formation pathways. While in winter, the dominant $N_2O_5$ hydrolysis pathway



produces nitrate with high $\Delta^{17}O$. Such seasonal patterns have been widely observed globally as summarized in Alexander et al. (2020).

The modeled $\delta^{15}N$ of $F_{pri}$ ranges from −10.3 ‰ to 5.0 ‰ which falls well within the reported atmospheric $\delta^{15}N(NO_3^-)$ values in continental and marine boundary layer in both hemispheres in regions not impacted by snowpack emission (Li et al., 2022; Lim et al., 2022; Morin et al., 2009; Shi et al., 2021). However, the seasonal pattern of $\delta^{15}N$ of $F_{pri}$ which displays a summer maximum is opposite to the typical seasonal

pattern of atmospheric $\delta^{15}N(NO_3^-)$ found in mid-latitude continental areas, where higher $\delta^{15}N(NO_3^-)$ values in winter and lower $\delta^{15}N(NO_3^-)$ values in summer are widely observed (e.g., Beyn et al., 2014; Freyer, 1991; Fang et al., 2021; Lim et al., 2022; Esquivel Hernández et al., 2022). This summer high and winter low $\delta^{15}N(NO_3^-)$ in $F_{pri}$ is instead consistent with the observations at two Arctic coastal sites (Morin et al., 2012;

Morin et al., 2008), where the summer high atmospheric $\delta^{15}N(NO_3^-)$ is strongly correlated with air temperature. Morin et al. (2008) suggest the $\delta^{15}N(NO_3^-)$-temperature relationship observed at the Arctic coastal sites may be related to physicochemical transformations of nitrate in Arctic and during the transport of nitrate and reactive nitrogens from the mid-latitudes, though the specific mechsmimes is unkonwn.

Another possibility to explain the modeled summer higher $\delta^{15}N(NO_3^-)$ in $F_{pri}$ is that there may be more anthropogenic nitrate transported from mid-latitudes to Greenland in summer than in winter. $F_{pri}$ is comprised of nitrate originating from the mid-latitudes as well as nitrate formed in the Arctic region. Morin et al. (2009) suggested that air parcels originating from regions with more anthropogenic impacts

carries nitrate with higher $\delta^{15}N$, which was confirmed by subsequent studies (Li et al., 2022; Vicars and Savarino, 2014; Shi et al., 2021). The increased frequency of air sources originating from the North America in summer compared to winter (Kahl et al., 1997) could thus lead to more anthropogenic nitrate to Greenland in summer, resulting in higher summer $\delta^{15}N$ of primary nitrate than winter.

The potential link between $\delta^{15}N$ of $F_{pri}$ and its precursor $NO_x$ emissions is not further discussed here, as recent studies have shown that the isotopic effect during $NO_x$





photo-recycling is complex (Li et al., 2020) and may dominate $\delta^{15}$N variations in atmospheric nitrate (Fang et al., 2021; Li et al., 2021). More comprehensive studies on the isotopic effects during atmospheric nitrate formation as well as the potential

fractionation during transport are required to further link $\delta^{15}$N of F$_{pri}$ with its precursors and/or source regions.

### 4.2 Model results at Dome C, Antarctica

### 4.2.1 Snowpack nitrate profile in the photic zone

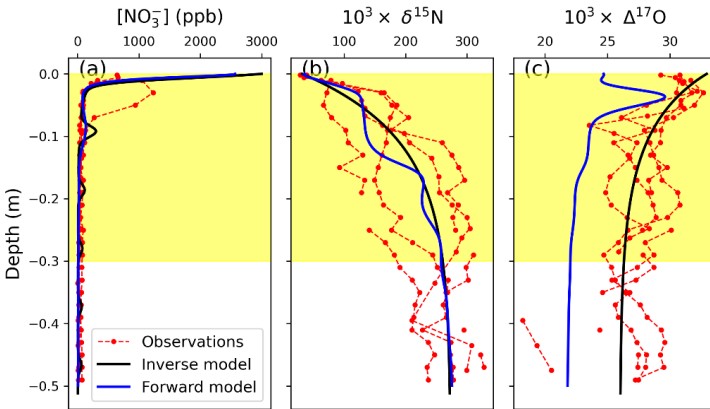

**Figure 5.** Comparison between the observed and modeled snowpack nitrate concentrations, $\delta^{15}$N, and $\Delta^{17}$O at Dome C. The red lines with circles represent four observed snowpack nitrate profiles at Dome C from Erbland et al. (2013) and Frey et al. (2009), while the blue and black lines are modeled results from the forward model (i.e., the TRANSITS model) and the inverse model in this study, respectively. The
yellow background represents the depth of the photic zone.

Since Dome C snowpack exhibits very distinct trends in the concentration and isotopic ratio of nitrate in the photic zone, we first examine the modeled summer snowpack nitrate profile at Dome C in comparison with the previous observations

(Erbland et al., 2013; Frey et al., 2009) in Fig. 5. The TRANSITS model results (Erbland et al., 2015) are also shown in Fig. 5 for comparison. Both models reproduce the observed decrease in nitrate concentration and the large enrichments in $\delta^{15}$N(NO$_3^-$) well. We note that the predicted surface snow nitrate concentration is higher than the observations by both models. This is because the modeled concentration represents the





state that atmospheric nitrate has just deposited onto the snow surface, while the observed skin layer snow may have already undergone snow metamorphism and/or post-depositional processing (Winton et al., 2020). This is also supported by recent observations at Dome C that newly deposited diamond dust could possess nitrate concentrations up to 2000 ppb (Winton et al., 2020), within the range of model

predictions.

The decreasing trend in $\Delta^{17}O(NO_3^-)$ within the photic zone is also reproduced by these two models, caused by the cage effect during nitrate photolysis. However, the TRANSITS model appears to underestimate snowpack $\Delta^{17}O(NO_3^-)$ while the inverse model performs better in snowpack $\Delta^{17}O(NO_3^-)$ simulation. This is because in the

TRANSITS model, snow $\Delta^{17}O(NO_3^-)$ is controlled by a combination of $\Delta^{17}O(NO_3^-)$ of FD and the subsequent cage effect after deposition. At Dome C, $\Delta^{17}O(NO_3^-)$ of FD is dominated by locally formed atmospheric nitrate (i.e., FP) (Erbland et al., 2015), which is in turn determined by the prescribed $\Delta^{17}O$ transfer during NO-NO$_2$ cycling and the subsequent OH oxidation of NO$_2$ under sunlight conditions in the model. However,

Savarino et al. (2016) demonstrated that the standard chemistry (i.e., exclusive oxidation of NO$_2$ by OH in summer) and the associated isotopic mass balance applied to $\Delta^{17}O$ (i.e., the one used by the direct model) does not hold in the Dome C atmosphere, with this standard approach systematically underestimating the observations. Our inverse model is in line with this conclusion. The inverse model calculates atmospheric

$\Delta^{17}O(NO_3^-)$ from the archived snow $\Delta^{17}O(NO_3^-)$ by subtracting the cage effect but does not assume any specific chemical reaction in the atmospheric box, contrary to the forward model. Therefore, although the inverse model uses the same method as the TRANSITS model to calculate $\Delta^{17}O(NO_3^-)$ of locally formed atmospheric nitrate (FP), it does not include any hypothesis of how local nitrate is formed. The good match

between observations and inverse model output is a further demonstration that atmospheric $\Delta^{17}O(NO_3^-)$ is not in agreement with the standard daylight chemistry of nitrate formation (i.e. NO$_2$+OH).

Overall, the consistency of the modeled and observed snowpack nitrate profiles





suggests that the effect of post-depositional processing is properly represented by the

inverse model. This confirms that the inverse model can properly reproduce snow

nitrate concentrations and isotopes in the photic zone, which are intermediate status

between archived and atmospheric nitrate.

### 4.2.2 Skin layer and atmospheric nitrate

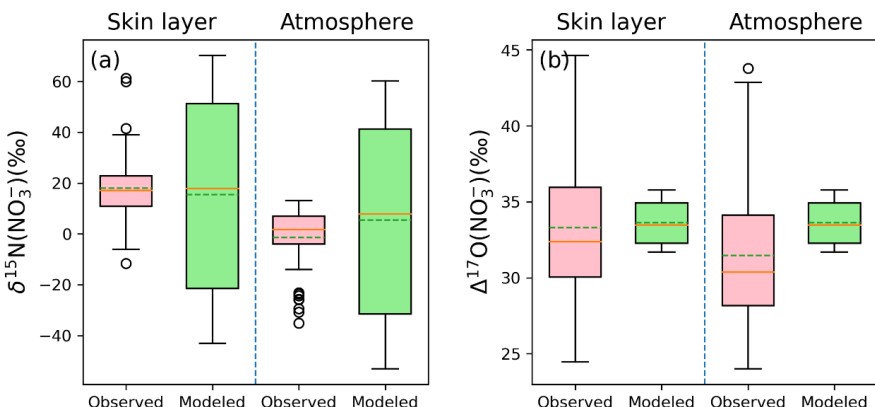

**Figure 6.** Comparison between the observed and modeled annual averages of
$\delta^{15}N(NO_3^-)$ and $\Delta^{17}O(NO_3^-)$ in the atmosphere and snow skin layer at Dome C. The
solid line in the box plot indicates the median value, while the dash line represents the
average value.

For Dome C, since the seasonal information of the archived nitrate profiles is
unknown, although the time step in the model was set to one week, we mainly focus on

the annual average values. We note that the modeled isotopic compositions of snowpack

and skin layer nitrate are irrelevant to the prescribed nitrate concentration seasonality.

This is because the total nitrate loss fraction and the induced isotopic effect only depend

on the total amount of actinic flux received during snow burial. In the following

discussion we only report and discuss the modeled isotopes of local atmospheric and

skin layer nitrate from case 1, i.e., the archived snow nitrate concentration was assumed

to be constant throughout the year.

The observed annual average $\delta^{15}N(NO_3^-)$ and $\Delta^{17}O(NO_3^-)$ values in the skin layer
at Dome C are $18.0 \pm 11.7$ ‰ and $33.6 \pm 1.4$ ‰, respectively (Erbland et al., 2013),

while the modeled skin-layer values are $15.7 \pm 38.6$ ‰ and $33.3 \pm 4.7$ ‰, respectively,





in good agreement with the observations. The observed annual average atmospheric $\delta^{15}N(NO_3^-)$ and $\Delta^{17}O(NO_3^-)$ values are -1.3 ± 11.6 ‰ and 31.4 ± 4.6 ‰, respectively, while the modeled values are 8.0 ± 11.7 ‰ and 33.6 ± 1.4 ‰, respectively. Note the average observed $\delta^{15}N$ values in this study were calculated as arithmetic mean instead of mass-weighted mean reported in Erbland et al. (2013) since the inverse model cannot directly calculate the nitrate concentration in the atmosphere. Nevertheless, the modeled averages are similar to the observed averages except for atmospheric $\delta^{15}N(NO_3^-)$. The difference between the modeled and observed atmospheric $\delta^{15}N(NO_3^-)$ could be again related to constant $\varepsilon_d$ used in the model. As discussed earlier, in the model we followed Erbland et al. (2015) to set $\varepsilon_d$ = 10 ‰ throughout the year, while observations at Dome C indicate that $\varepsilon_d$ could be as large as 25 ‰ in summer instead of 10 ‰ (Erbland et al., 2013). Hence the modeled atmospheric $\delta^{15}N(NO_3^-)$ could be overestimated. This reinforces that it is necessary to further explore the isotope effects on $\delta^{15}N(NO_3^-)$ during atmospheric nitrate deposition.

### 4.2.3 Flux and isotopes of primary nitrate

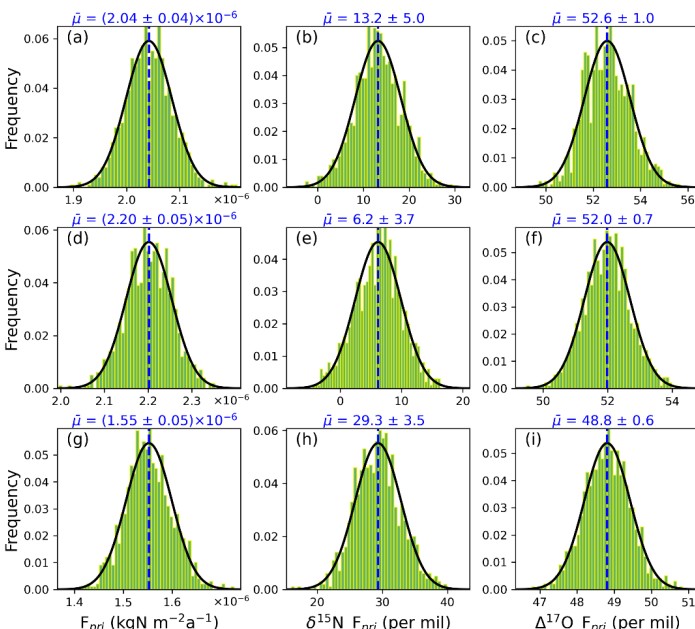

**Figure 7.** Frequency histogram of the calculated primary nitrate flux and its mean



$\delta^{15}$N/$\Delta^{17}$O values at Dome C under three different archival snow nitrate concentration distributions: (**a-c**) Case 1: uniform distribution, (**d-f**) Case 2: Gaussian-type function, (**g-i**) Case 3: shifted Gaussian-type function. The black solid lines represent the fitted Gaussian function of the frequency distribution. The blue dashed lines represent the mean values of $F_{pri}$ and its $\delta^{15}$N and $\Delta^{17}$O, which are labeled on the top of each subplot.

In Fig. 7, the flux of primary nitrate ($F_{pri}$) and its mean isotopes from the 3 difference cases (i.e., different nitrate concentration seasonality in archived snow) are displayed. Similar to the previous section, we only focus on their annual means. Note when calculating $\Delta^{17}$O(NO$_3^-$) of $F_{pri}$, $\Delta^{17}$O values of locally formed nitrate are necessary. As discussed in 4.2.1 that following the same method in the TRANSITS model to calculate $\Delta^{17}$O(NO$_3^-$) of locally formed atmospheric nitrate would underestimate $\Delta^{17}$O(NO$_3^-$) of FP at Dome C. This is especially evident in summer when the snow sourced nitrate (i.e., FP) dominates the atmospheric nitrate budget, and the calculated $\Delta^{17}$O(NO$_3^-$) of FP by the TRANSITS model is about 6 ‰ lower than observed atmospheric $\Delta^{17}$O(NO$_3^-$) (Erbland et al., 2015). Thus, in the inverse model, when calculating $\Delta^{17}$O(NO$_3^-$) of $F_{pri}$ at Dome C, $\Delta^{17}$O(NO$_3^-$) of FP was not calculated using the model default method as in the TRANSITS, but prescribed as the observed atmospheric summer $\Delta^{17}$O(NO$_3^-$). Otherwise, the modeled $\Delta^{17}$O(NO$_3^-$) of $F_{pri}$ would be higher than 70 ‰, which is highly unrealistic. Note, this is not an issue at Summit Greenland because FP doesn't dominate the atmospheric nitrate budget in summer there.

As shown in Fig. 7, although the prescribed archived nitrate concentration seasonality does not alter the modeled snowpack and atmospheric nitrate isotopes, it has a profound impact on the modeled primary nitrate flux and its isotopes. In particular, under the three cases of different seasonal distributions of the archived snow nitrate concentrations, the modeled $F_{pri}$ and its annual mean $\delta^{15}$N and $\Delta^{17}$O range from 1.5 to 2.2 ×10$^{-6}$ kgN m$^{-2}$ a$^{-1}$, 6.2 to 29.3 ‰ and 48.8 to 52.6 ‰, respectively. The inverse model calculated $F_{pri}$ is smaller than the value used in the original TRANSITS (8.2×10$^{-6}$ kgN m$^{-2}$ a$^{-1}$) in Erbland et al. (2015), but this is easily resolved given the large uncertainty in the archived nitrate concentration used as model input.

The modeled annual mean $\delta^{15}$N of $F_{pri}$ ranges from 6.2-29.3 ‰, in contrast with the





observed atmospheric $\delta^{15}N(NO_3^-)$ in southern mid-latitude area or the Southern Ocean where $\delta^{15}N(NO_3^-)$ is in general negative or close to 0 (Morin et al., 2009; Shi et al., 2018; Shi et al., 2021). The modeled positive $\delta^{15}N$ of $F_{pri}$ is however consistent with the wintertime atmospheric $\delta^{15}N(NO_3^-)$ observed in Antarctica when the effect of photolysis is null and local atmospheric nitrate likely reflects $F_{pri}$. The maximum atmospheric $\delta^{15}N(NO_3^-)$ in winter was found to be 10.8 ‰ at DDU (Savarino et al., 2007), 12.8 ‰ at Dome C (Erbland et al., 2013) and 13.9 ‰ at Zhongshan station (Shi et al., 2022). These positive $\delta^{15}N$ values have been link to stratospheric denitrification as nitrate produced in stratosphere is suggested to be 19 ± 3 ‰ by considering the fractionation induced by different of $N_2O$ photolysis channels (Savarino et al., 2007). Therefore, the modeled flux and $\delta^{15}N$ of $F_{pri}$ points towards the dominance of stratospheric denitrification in nitrate budget at Dome C.

The modeled $\Delta^{17}O$ of $F_{pri}$ is also very high for all three cases (48.8-52.6 ‰). The measured bulk $\Delta^{17}O$ of surface ozone in Antarctica is about 26 ‰ (Ishino et al., 2017; Savarino et al., 2015) that fits well with the global tropospheric average of 25.4 ‰ (Vicars and Savarino, 2014). Given that the oxygen mass-independent fractionation signal of ozone is mainly occupied by the terminal oxygen atom and transferred to other molecular, atmospheric nitrate of tropospheric origin should possess a $\Delta^{17}O$ signal less than 39 ‰ (Mauersberger et al., 2003; Savarino et al., 2008), which cannot explain our calculated high $\Delta^{17}O$ of $F_{pri}$. However, the bulk $\Delta^{17}O$ of stratospheric ozone was measured to be 34.3 ± 3.6 ‰ (Lämmerzahl et al., 2002; Krankowsky et al., 2000), which indicated that nitrate produced in the stratosphere could gain a higher $\Delta^{17}O$ signature from ozone (Lyons, 2001). It has been observed in Antarctica that the atmospheric $\Delta^{17}O(NO_3^-)$ could exceed 40 ‰ in winter and early spring when stratospheric denitrification occurs (Ishino et al., 2017; Walters et al., 2019; Erbland et al., 2013; Savarino et al., 2007; Shi et al., 2022). A recent study also revealed that the surface snow $\Delta^{17}O(NO_3^-)$ at Dome C frequently exceeds 40 ‰ during winter/spring and could sometimes reach up to 50 ‰ (Akers et al., 2022). As we mentioned previously, these winter $\Delta^{17}O(NO_3^-)$ observations likely reflect the primary nitrate signal at that time





since the photolysis of snow nitrate does not occur due to lack of sunlight. Thus, the high modeled $\Delta^{17}O$ of $F_{pri}$ seems to again indicate a dominant role of stratosphere denitrification in external nitrate source to Dome C, similar to what can be reflected from the modeled $\delta^{15}N$ of primary nitrate. In addition, Erbland et al (2015) estimated that stratospheric denitrification nitrate flux is $(4.1 \pm 2.5) \times 10^{-6}$ kgN m$^{-2}$ a$^{-1}$ in Antarctica,

while our calculated $F_{pri}$ of 1.5 - 2.2 $\times 10^{-6}$ kgN m$^{-2}$ a$^{-1}$ at Dome C is within the same range.

    Finally, we acknowledge that there are many factors that would affect the model results, such as the archived snow nitrate concentration and isotopes, the export fraction ($f_{exp}$), and the cage effect fraction ($f_c$). These need to be further explored by observations

to improve the model performance.

## 5. Model sensitivity tests: the impact of $f_{exp}$ and $f_c$

    In this section, we report the sensitivity test results to elucidate the impact of two model parameters that lack of direct observational constraints, the export fraction ($f_{exp}$) and the cage effect fraction ($f_c$). We mainly focus on the annual net loss and the

differences in isotopes between $F_{pri}$ and FA in accordance with the resolution of ice core measurements.

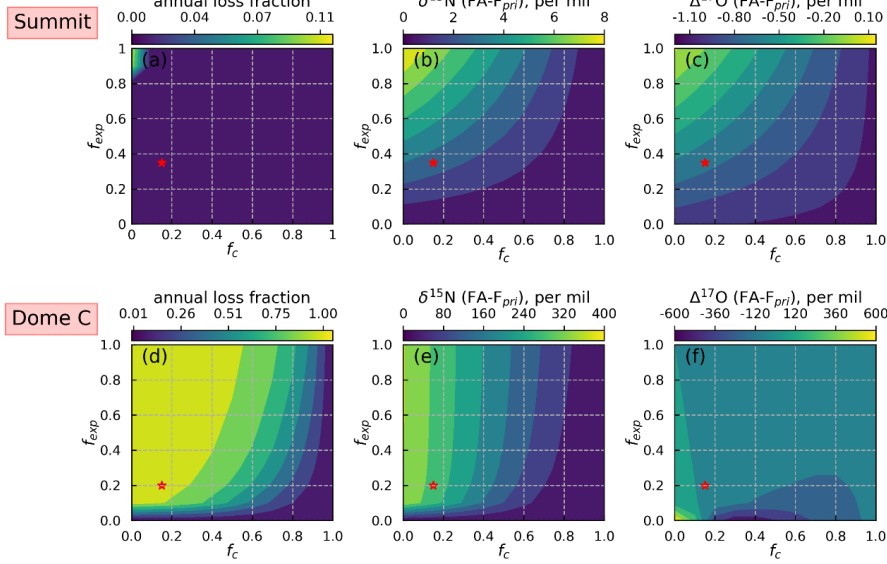



**Figure 8.** Model sensitivity test results of two parameters $f_{exp}$ and $f_c$ for Summit (**a-c**) and Dome C (**d-f**). The annual nitrate loss fraction is defined as 1-FA/F$_{pri}$ following Jiang et al. (2021). Red stars represent the values of $f_{exp}$ and $f_c$ used in model simulations.

The sensitivity test results are shown in Fig 8. The annual loss fraction (defined as 1-FA/F$_{pri}$) represents the final preservation of primary nitrate after post-depositional processing (Jiang et al., 2021). The inverse model predicts an annual loss fraction of 3.5% under present Summit conditions, which is close to the TRANSITS model prediction of 4.1% (Jiang et al., 2021). This small discrepancy is likely caused by the use of simplified snow radiative transfer parameterization in the inverse model. In addition, the differences of $\delta^{15}$N and $\Delta^{17}$O between FA and F$_{pri}$ are also in good agreement with the TRANSITS model. As expected, larger $f_{exp}$ and smaller $f_c$ would result in a higher degree of net loss in F$_{pri}$ and larger isotopic effects in both $\delta^{15}$N and $\Delta^{17}$O. However, under present-day conditions at Summit, the preserved snow nitrate concentrations and isotopes at the annual scale is only altered slightly and the degree of changes is insensitive to $f_{exp}$ and $f_c$.

For Dome C, the model results are sensitive to $f_{exp}$ when $f_c$ is small, and becomes sensitive to $f_c$ when $f_{exp}$ is larger. In addition, the $\Delta^{17}$O results display a non-monotonic response to these two parameters, especially when $f_{exp}$ approaches zero (Fig. 8f). A similar phenomenon was seen in the TRANSITS model simulations in Erbland et al. (2015), where they found that the model results could not converge when $f_{exp}$ was set to zero. The high sensitivity of model parameters renders difficult to reconstruct the historical variations in primary nitrate based on ice core records at Dome C unless these parameters are precisely constrained. For present day conditions, $f_{exp}$ and $f_c$ could be constrained by atmospheric and snowpack observations (Erbland et al., 2015) but it is unknown if these values could be applied to different climate conditions. In addition, the difficulties in choosing an appropriate archival nitrate concentration profile as model initial conditions would add extra uncertainties to the model results.

**6. Conclusions and implications**

In this study, we introduce an inverse model which is designed to correct for the



effects of post-depositional processing on ice-core nitrate concentration and its isotopes.
The model was tested against present-day Summit, Greenland and Dome C, Antarctica conditions to validate its performance under different snow accumulation rates. Model results compared to observations demonstrate that the inverse model is capable of adequately correcting the effect of post-depositional processing. The modeled atmospheric nitrate $\delta^{15}N/\Delta^{17}O$ at Summit are generally in good agreement with observations but with slight underestimate in winter $\delta^{15}N(NO_3^-)$, which is likely because the model doesn't treat the correctly the seasonal differences in nitrogen isotope fractionation during deposition ($\varepsilon_d$). At Dome C, the model also well reproduced the observed snowpack nitrate profiles in the photic zone, the annual skin layer $\delta^{15}N/\Delta^{17}O(NO_3^-)$, and atmospheric $\Delta^{17}O(NO_3^-)$ at Dome C, but again overestimated the average atmospheric $\delta^{15}N(NO_3^-)$ probably also due a low bias in $\varepsilon_d$ used in the model. A better quantification on the isotope fractionation of $\delta^{15}N(NO_3^-)$ during deposition is therefore needed.

The inverse working flow of this new model also enables us to qualitatively retrieve information regarding primary nitrate from the archived snow nitrate. The calculated seasonality in $\delta^{15}N$ of $F_{pri}$ at Summit displays a maximum in mid-summer that is distinct from the observed spring $\delta^{15}N(NO_3^-)$ peak in snowpack. This seasonal pattern is in contrast with observed atmospheric $\delta^{15}N(NO_3^-)$ variations in mid-latitudes which is thought to be the major aerosol source region to Summit, but is consistent with the atmospheric $\delta^{15}N(NO_3^-)$ variations observed in the high-latitude Arctic region. The $\delta^{15}N$ of $F_{pri}$ may reflect seasonally-varied main source regions to Greenland or a dominate role of high-latitude nitrate transport to Summit. At Dome C, both the magnitude of $F_{pri}$ and its $\delta^{15}N/\Delta^{17}O$ indicate a dominant role of stratospheric denitrification on nitrate budget at Dome C.

The inverse model is designed to help interpret ice core nitrate records. Applying the inverse model to ice core nitrate records needs knowledge of initial conditions. In particular, archived snow nitrate concentration and its $\delta^{15}N$ and $\Delta^{17}O$, the snow accumulation rate and light absorption impurity concentrations should be known for a





given ice core. In addition, chemistry-climate models such as the ICECAP or GCAP

model (Murray et al., 2014; Murray et al., 2021) would be also necessary to provide

extra constraints, such as the oxidizing agent concentrations, total column ozone (TCO),

wind field and boundary layer heights for the past climates and are required to estimate

$\Delta^{17}$O of FP and $f_{exp}$ (Alexander et al., 2020; Jiang et al., 2021). The calculated primary

nitrate flux and its $\delta^{15}$N and $\Delta^{17}$O can be further combined with the chemistry-climate

model results to interpret its climate implications such as the variations in tropospheric

$NO_x$ and oxidant abundance, which would improve our understanding of key factors

controlling the variability in atmospheric oxidation capacity under different climates.

**Appendix A: derivation of the nitrate mass and isotopic balance equations
during photolysis (Eq (7-9))**

In the inverse model, we follow Erbland et al. (2015) to separate the photolysis of

nitrate on ice grain into two steps, i.e., the direct photolysis followed by a subsequent

cage effect (Fig. A1). It is well-documented that secondary chemistry can occur during

snow nitrate photolysis to reform nitrate and alter the isotopes of the remaining nitrate,

which is termed as the cage effect (McCabe et al., 2005; Meusinger et al., 2014). To

quantify this effect, Erbland et al. (2015) assigned an empirical parameter ($f_c$) to

represent the fraction of the nitrate photoproduct that would undergo cage effect, and

derived a value of 0.15 for $f_c$ based on the observed decreasing trend of $\Delta^{17}$O ($NO_3^-$) in

snowpack at Dome C. Assuming a fraction ($f_p$) of initial snow nitrate was photolyzed

and undergone cage effect, the mass balance equation for snow nitrate can be written

as:

$$c(SN) = c(SN')\left(1 - f_p + f_c f_p\right) \tag{A1}$$

We follow the previous appoint that the superscript represents the state before being

photolyzed. It can be easily seen that Eq (A1) is equal to Eq (7).

The direct photolysis of snow nitrate obeys the Rayleigh equation with a nitrogen

fractionation constant of $\overline{\varepsilon_p}$. The $\delta^{15}$N of the photoproduct ($NO_2$) and the remaining

nitrate on snow ($SN_r$) can be approximated by:



$$\delta^{15}N(NO_2) \approx \delta^{15}N(SN') + \frac{\bar{\varepsilon_p}(1-f_p)ln(1-f_p)}{f_p} \tag{A2}$$

$$\delta^{15}N(SN_r) \approx \delta^{15}N(SN') - \bar{\varepsilon_p}ln(1-f_p) \tag{A3}$$

Here we use the mathematic approximation of $ln(1+x) \sim x$. The $\delta^{15}N$ of the state after photolysis can be calculated via isotopic mass balance equation:

$$\delta^{15}N(SN) = \frac{(1-f_p)\delta^{15}N(SN_r) + f_c f_p \delta^{15}N(NO_2)}{1-f_p+f_c f_p}$$

$$= \delta^{15}N(SN') + \frac{(1-f_p)(1-f_c)\bar{\varepsilon_p}\ln(1-f_p)}{(1-f_p)+f_c f_p} \tag{A4}$$

which is equal to Eq (8).

For $\Delta^{17}O$, it is assumed that direct photolysis of nitrate will only induce mass-dependent fractionation and has no impact on $\Delta^{17}O$. However, the produced $NO_2$ which is re-oxidized by OH radical would lower its $\Delta^{17}O$ by 2/3 since OH radical rapidly archives isotope equilibrium with the surrounding water molecule and erase its $\Delta^{17}O$ signal. Hence, by again using the isotopic mass balance equation, it can be shown that:

$$\Delta^{17}O(SN) = \frac{(1-f_p)\Delta^{17}O(SN') + \frac{2}{3}f_c f_p \Delta^{17}O(SN')}{1-f_p+f_c f_p}$$

$$= \frac{1-f_p+\frac{2}{3}f_c f_p}{1-f_p+f_c f_p}\Delta^{17}O(SN') \tag{A5}$$

which is equal to Eq (9).

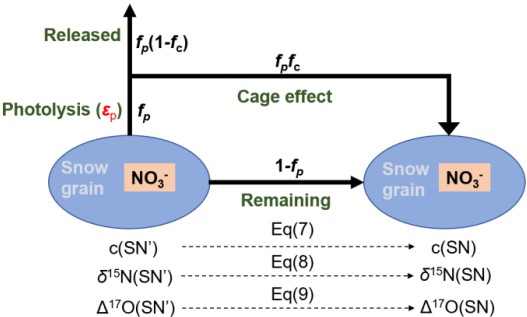

**Figure A1.** Sketch of the mass and isotopic transfer relationship during nitrate photolysis on snow grains. The black italic variables near the arrows indicate the fractional change in each subprocess.

**Appendix B: method for calculating $\Delta^{17}O(NO_3^-)$ of FP**





The $\Delta^{17}O(NO_3^-)$ of FP is required to solve the mass balance equations. We follow

the standard algorithm to calculate atmospheric $\Delta^{17}O(NO_3^-)$ that has been widely used

by previous studies (Alexander et al., 2020 and reference within). Atmospheric $NO_2$ is

assumed to rapidly achieve photochemical steady state (PSS) so that its $\Delta^{17}O$ can be

represented as follow:

$$\Delta^{17}O(NO_2) = \frac{k_{NO+O_3}[O_3] + k_{NO+BrO}[BrO]}{k_{NO+O_3}[O_3] + k_{NO+BrO}[BrO] + k_{NO+RO_2}[RO_2]} \Delta^{17}O(O_3^*) \quad \text{(B1)}$$

where $k$ represents different oxidation channel for NO in atmosphere and $\Delta^{17}O(O_3^*)$

represents $\Delta^{17}O$ of the terminal oxygen in ozone molecule, and $RO_2$ includes both $HO_2$

and other organic peroxyl radicals. Thus, to obtain $\Delta^{17}O(NO_2)$, concentrations of ozone

and oxidizing radicals are necessary. For the subsequent oxidation of $NO_2$, only

$NO_2$+OH channel is considered, and $\Delta^{17}O$ of OH is assumed to be zero owing to its

equilibrium with atmospheric $H_2O$. Thus, $\Delta^{17}O$ of the locally formed atmospheric

nitrate (i.e., FP) can be calculated by:

$$\Delta^{17}O(NO_3^-) = \frac{2}{3}\Delta^{17}O(NO_2) \quad \text{(B2)}$$

**Appendix C: adjusting the photolysis quantum yield used in Dome C simulations**

The photolysis quantum yield ($\Phi$) previously used in TRANSITS model simulation

at Dome C was set to 0.026 in Erbland et al. (2015), which was adjusted to 0.015 in this

study. Both values are obtained by best fitting the observed snowpack nitrate

concentration and its $\delta^{15}N$ profiles with model output, and are within the range of

measured quantum yield (0.003-0.44) of Dome C snow nitrate (Meusinger et al., 2014).

However, using a value of 0.026 in the inverse model would largely overestimate the

photolytic loss of snow nitrate, resulting in unrealistically high nitrate concentration in

skin layer (>15000 ppb) and excessive fractionation in $\delta^{15}N(NO_3^-)$ (500 ‰) and

$\Delta^{17}O(NO_3^-)$ (-18 ‰). Adjusting the quantum yield to 0.015 could well reproduce the

observed nitrate concentration and $\delta^{15}N(NO_3^-)$ in skin layer and snowpack.

The discrepancy in the chosen quantum yield between these two models is caused

by whether the diffusion process of snow nitrate is included, as diffusion would tend to

smooth the entire snowpack nitrate profiles and decrease the asymptotic values



(Erbland et al., 2015). The omission of the diffusion process in the inverse model is based on the following considerations. First, the snowpack nitrate profile at sites with even lower accumulation rates (Dome A in East Antarctica, with accumulation rate of

~ 19 kg m$^{-2}$ a$^{-1}$) does not display detectable smoothing effect on snowpack nitrate or its $\delta^{15}$N and $\Delta^{17}$O (Shi et al., 2015), suggesting that diffusion is not as important as assumed in the TRANSIST model. Second, the TRANSIST model would induce numerical diffusion during the division of snow layers in each time step, which results in rapid decreases in the amplitude of the simulated seasonality in snow nitrate, as can be seen

in the simulated snowpack profiles in Winton et al. (2020).

**Appendix D: fitting the shape parameters of skin layer nitrate concentration distributions at Dome C**

To obtain a hypothetical seasonal pattern in the archived nitrate concentration, we assume that the normalized archived nitrate concentrations (by its arithmetic mean)

would follow a same Gaussian-type distribution as nitrate concentrations in the skin layer. The shape parameters (a, b, σ) of the Gaussian distribution are determined by using the least square regression method.

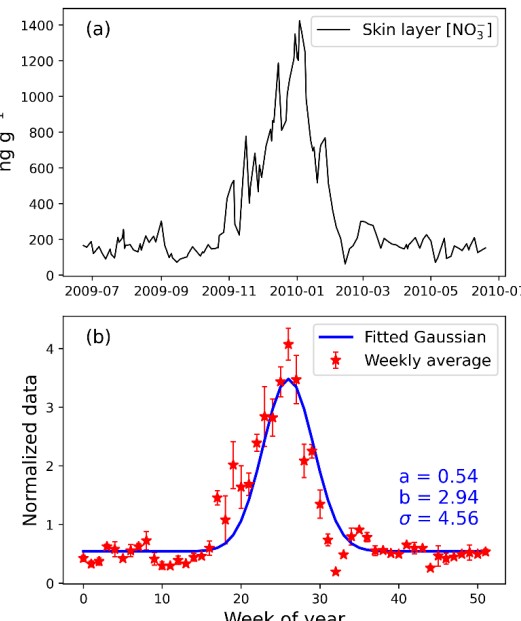

**Figure D1. (a)** Observed annual variations of skin layer nitrate concentration at Dome



C (Erbland et al., 2013). (**b**) Normalized skin layer nitrate concentration by its arithmetic mean (red star) and the fitted Gaussian curve, with the shape parameters (a, b, σ) labeled in blue text.

*Code/data availability.* The codes for the numerical simulations and their analysis will

be provided upon direct request to the corresponding author.

*Author contributions.* L.G conceived this study. Z.J. wrote the code, performed the model simulations, analyzed the data, and wrote the manuscript with L.G. J. S. and B.A. provided suggestions for model development and result interpretation. All authors gave

feedback on the paper writing.

*Competing interests.* The contact author has declared that none of the authors has any competing interests.

**Acknowledgements:**

L.G. acknowledges financial support from the National Natural Science Foundation of China (Awards: 41822605 and 41871051) and the Strategic Priority Research Program of Chinese Academy of Sciences (XDB 41000000), and the National Key R&D Program of China (2022YFC3700701). This work was partially supported

by the French national programme LEFE/INSU (IMAGO), the ANR grants ANR-15-IDEX-02 (project IDEX Université Grenoble Alpes) and Labex OSUG@2020 (Investissements d'avenir – ANR10 LABX56), the support of the overwintering staffs and the French polar institute IPEV through the SUNNITEDC (1011) and CAPOXI programs (1177) (J.S.). B. A. acknowledges support from NSF PLR 1542723 and AGS

2202287 and 1702266.

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
