# Peer review of "An inverse model to correct for the effects of postdepositional processing on ice-core nitrate and its isotopes: model framework and applications at Summit, Greenland and Dome C, Antarctica"

_EGUsphere, 2023_

## Referee Comment (RC2)

[referee-annotated manuscript omitted]

---

## Author Comment (AC1)

We are grateful to the reviewer for the valuable suggestions and/or comments which improve the manuscript significantly. Below we list the detailed responses to the reviewer's suggestions and comments. The comments are listed in italics, followed by the response in normal font with changes highlighted in blue.

**Response to Referee #1**

*Ice core nitrate isotope compositions may be useful to reconstruct atmospheric nitrate isotope compositions in the past with paleoclimate implications. Previous studies have investigated impacts of post-depositional processing on isotope compositions of nitrate preserved in snow and ice. In this work, Jiang et al. revised the TRANSITS model (a one-dimension snow photochemistry model) to calculate atmospheric nitrate deposition flux and isotope compositions based on snow records. Exemplary applications were applied to Summit and Dome C data and the calculated results were compared with measurement data. Although future efforts are needed to further evaluate and improve this model, this work is an important step towards a more precise interpretation of ice core nitrate isotope data. The equation derivation appears correct, and the model logic looks scientifically reasonable to me. However, the writing of this manuscript is too technical to readers outside the small community of post-depositional possessing of nitrate isotopes, and one may need to read all papers written by the authors previously to understand this work. This writing style is not easy for casual readers (especially for atmospheric scientists who do not work in cryospheric sciences and isotopes) to follow. I spent considerable time to digest the manuscript, even though I am kind of familiar with topics discussed in this manuscript. I therefore have some suggestions that aim to improve the clarity of this manuscript.*

**Response:** We really appreciate the reviewer's time and effort on evaluating the manuscript, and thanks for the positive comments. Indeed, the topic of the manuscript is for a relatively small community and would be difficult to understand without sufficient background. In the revised manuscript, we have followed the reviewer's suggestion and try to improve the readability of the manuscript.

*Line 29: Please define $F_{pri}$ at the very beginning. For readers who did not read the authors' previous papers, they would not understand what it is.*
**Response:** We have added the following text at the beginning of the abstract:
"…reconstruct primary nitrate flux (i.e., the deposition flux of nitrate to surface snow that originates from long-range transport or stratospheric input) and its isotopes…"

*Lines 263-265: Please rewrite this sentence. It is not clear.*
**Response:** The original sentence "…The subsequent conversion of $NO_2$ to nitrate would also determine 1/3 of the oxygen atom of the newly formed nitrate." has been rephrased as follow:
"…During the subsequent oxidation of atmospheric $NO_2$, one more oxygen atom inherited from the oxidants (e.g., OH or BrO) is incorporated into one newly formed

HNO$_3$ molecule. Thus, $\Delta^{17}O(FP)$ can be represented by 2/3 of $\Delta^{17}O(NO_2)$ plus 1/3 of $\Delta^{17}O(oxidant)$"

*The definition listed in Table 1 is unclear. For example, I do not understand what "d15N of archived nitrate flux" means. Does a flux have a d15N value? Does it mean d15N of archived nitrate? In my opinion, these definitions make the manuscript difficult to follow.*

**Response:** Thank you for pointing this out. We follow the reviewer's suggestion to avoid the term "$\delta^{15}N$ of xx nitrate flux" and substitute it to "$\delta^{15}N$ of xx nitrate" in the revised manuscript. In this case, "$\delta^{15}N$ of archived nitrate flux" was revised as "$\delta^{15}N$ of archived nitrate".

*There are 13 input parameters listed in Table 1, but only 6 are described in Table 2. It is difficult for readers to check everything throughout different parts of the manuscript and from different papers.*

**Response:** Thanks for this point. We have added the omitted parameters in Table 2 to make it accordant with Table 1. The new Table 2 are shown as follows:

**Table 2.** Values of major parameter used in the model simulations at two different sites.

| Parameter | Dome C, Antarctica | | Summit, Greenland | |
| --- | --- | --- | --- | --- |
| | Value | Reference | Value | Reference |
| FA | $1.3\times10^{-7}$ kgN m$^{-2}$ a$^{-1}$ | Erbalnd et al. (2013) | $6.7\times10^{-6}$ kgN m$^{-2}$ a$^{-1}$ | Jiang et al., (2022) |
| $\delta^{15}N(FA)$ | 273.6 ‰ | Erbalnd et al. (2013) | 0.6 ‰ | Jiang et al., (2022) |
| $\Delta^{17}O(FA)$ | 26.0 ‰ | Erbalnd et al. (2013) | 27.9 ‰ | Jiang et al., (2022) |
| $A$ | 28 kg m$^{-2}$ a$^{-1}$ | Erbland et al. (2013) | 250 kg m$^{-2}$ a$^{-1}$ | Dibb et al., (2014) |
| $\rho$ | 300 | Erbland et al. (2013) | 380 | Geng et al. (2014) |
| TCO | 175-300 DU | Erbland et al. (2015) | 228-494 DU | Jiang et al., (2021) |
| $\Phi$ | 0.015 | Adjusted[a] | 0.002 | Jiang et al., (2021) |
| $\sigma$ | Wavelength dependent | Berhanu et al. (2014) | Wavelength dependent | Berhanu et al. (2014) |
| $\varepsilon_d$ | +10 ‰ | Erbland et al. (2013) | +10 ‰ | Erbland et al. (2013) |
| $\Delta^{17}O(NO_3^-)$ of FP | Observed atmospheric $\Delta^{17}O(NO_3^-)$ | Erbland et al. (2013) | Calculated | Jiang et al., (2021) |

| | | | | |
|---|---|---|---|---|
| $f_c$ | 0.15 | Erbland et al. (2015) | 0.15 | Erbland et al. (2015) |
| $f_{exp}$ | 0.2 | Erbland et al. (2015) | 0.35 | Jiang et al., (2021) |

[a]Adjusted according to the best fit of snowpack nitrate $\delta^{15}N$ profile at Dome C (Appendix C).

*Section 3: It is recommended to explicitly describe what parameters were used to do the calculation and what parameters were used to compare with the model results at each site. For example, it is stated that weekly data from Jiang et al. (2022) were used (Section 3.1). I need to re-read Jiang et al. (2022) to understand what these data are. In addition, I do not fully understand how the authors tested their model results. Did they use the atmospheric nitrate data reported in Jiang et al. (2022) or the surface snow as described in page 12? Where are the data from?*

**Response:** Thanks for this suggestion. We have added more detailed explanations regarding the model input data (i.e., parameters used to do model calculations) and the observational data used to compare with the model outputs. In particular, for Summit, the snowpack nitrate concentration and isotope data with weekly resolution compiled in Jiang et al. (2022) were used as model input values (i.e., archived snow nitrate properties with weekly resolution). To compare with the model outputs, i.e., the model calculated atmospheric nitrate isotopes based on observed the snowpack data, we used the observed atmospheric nitrate isotopes at monthly resolution reported by Jiang et al. (2022). For Dome C, we mainly used the annual skin layer and atmospheric nitrate isotopic observations from Erbland et al. (2013) as observational constraints to compare with the model calculated skin layer and atmospheric nitrate isotopes, where the snowpack nitrate isotopes below the photic zone were used as model input values.

In the revised manuscript, in section 3.1, we added: "…The snowpack nitrate concentration and isotope data with weekly resolution at Summit compiled in Jiang et al. (2022) were used as initial model input values to represent the archived snow nitrate signals."

At the beginning of section 4.1.1, we rephrased the original sentence to: "Currently there are no skin layer nitrate isotope observations at Summit, so we used the monthly atmospheric nitrate isotopic data from aerosol observations at Summit reported by Jiang et al. (2022) to compare with the modeled atmospheric nitrate isotopic variations…"

*Lines 387-389 and 397-398: I would not say that the seasonality of modeled d15N agrees well with observation based on Figure 3. As noted by the authors, the model cannot capture the seasonal variation from September to April.*

**Response:** We have weakened our statement in the revised manuscript:

"…As shown in Fig. 3, the modeled seasonality in atmospheric $\Delta^{17}O(NO_3^-)$ generally agrees well with the observed seasonal variations, while for $\delta^{15}N(NO_3^-)$, the model

predicted a similar seasonality as the observations, though in the winter half year the model underestimated the absolute values in comparison with the observations…"

*Lines 428-430: Could the authors tune the epsilon value in the model, give a quick estimation what epsilon value may reproduce the observational data, and briefly discuss if this epsilon value is reasonable? This test should be straightforward.*
**Response:** Thanks for this suggestion. In principle, the modeled atmospheric $\delta^{15}N(NO_3^-)$ = modeled $\delta^{15}N(FD)$ - $\varepsilon_d$. In the model, the $\varepsilon_d$ was set as +10 ‰ and this leads underestimations of atmospheric $\delta^{15}N(NO_3^-)$ in winter months compared to observations. In the above equation, if we replaced the **modeled** atmospheric $\delta^{15}N(NO_3^-)$ with the **observed** atmospheric $\delta^{15}N(NO_3^-)$, then the $\varepsilon_d$ should be the epsilon values needed to reproduce the observations. In the following figure (Fig. 1), we plotted the epsilon values needed to reproduce the observations by applying $\varepsilon_d$ = modeled $\delta^{15}N(FD)$ - **observed** atmospheric $\delta^{15}N(NO_3^-)$. As shown in the figure, the $\varepsilon_d$ is generally close to the model default value of +10 ‰ during the summer half year, while in winter half year the $\varepsilon_d$ is lower than +10 ‰ (except in March and December). This is consistent with what we speculated in the manuscript, i.e., the $\varepsilon_d$ in winter is lower than that in summer and this is probably the reason leading to the model underestimation in winter months.

[Figure]

**Figure 1.** The calculated $\varepsilon_d$ reproducing the observed atmospheric $\delta^{15}N(NO_3^-)$ at Summit using $\varepsilon_d$ = modeled $\delta^{15}N(FD)$ - **observed** atmospheric $\delta^{15}N(NO_3^-)$.

*Figure 5: I am confused what the results of "inverse model" mean. The inverse model used measured isotope values as input parameters. So I guess the model "results" plotted in this graph are the isotope values of "deposition nitrate" calculated from the model or the model input (calculated averages of measured values?). Please clarify.*
**Response:** In this figure, we plotted the observed and modeled nitrate concentration and isotopes in the photic zone. The modeled results are from the TRANSIT model and the inverse model. The differences between the two modeled results are that the forward model (TRANSITS model) uses the prescribed **isotopes of the primary nitrate as model inputs** to calculate snow nitrate concentration and isotopes **in the photic zone and the archived layers**, while the inverse model uses **the archived**

**snowpack nitrate concentrations and isotopes** (i.e., observed snowpack data well below the photic zone) as model inputs to calculate snow nitrate concentration and isotopes **in the photic zone including surface snow, and those in the atmosphere**. This is saying, both models are capable of simulating the snowpack nitrate profiles in the photic zone, and these are what plotted in Figure 5.

To make it more clear, in the revised manuscript, we have added the following statements at the end of section 2.3:

"…The archived nitrate profile could be dated by using various types of seasonal markers, such as the $\delta^{18}O$ of $H_2O$, the ion concentrations or their ratios, and the snow accumulation rates (Hastings et al., 2004; Furukawa et al., 2017; Dibb et al., 2007). As long as the archived snow nitrate profiles (i.e., snow nitrate concentration and isotopes below the photic zone) are given, the model can calculate nitrate concentrations and isotopes throughout the photic zone, and those in the atmosphere. The latter is considered as the atmospheric signals before being affected by post-depositional processing."

*Figure 6: Is it possible to show a similar figure for Summit so that readers can better understand how the model behaves if we just look at the annual average?*

**Response:** This is a good suggestion. But unfortunately, at Summit there is no skin layer observations so we can only compare the modeled and observed atmospheric values. The results are presented below as Fig.2. Overall, the modeled annual average values are in good agreement with the observations. For $\delta^{15}N(NO_3^-)$, the modeled and observed average values are -17.5 ± 3.0 ‰ and -14.8 ± 7.3 ‰ respectively, while for $\Delta^{17}O(NO_3^-)$ the values are 28.8 ± 2.6 ‰ and 28.6 ± 3.2 ‰. The small departure in $\delta^{15}N(NO_3^-)$ are likely caused by the assumed constant depositional fractionation factor used in the model as had been intensively discussed in the main text. Nevertheless, we think it's a good idea to provide comparison on the annual average values at Summit and we have added the following text in the revised version (but we didn't add the figure):

"…which is close to the value of 0.19 ‰ predicted by the TRANSITS model (Jiang et al., 2021). At an annual scale, the modeled and observed average atmospheric $\delta^{15}N(NO_3^-)$ are -17.5 ± 3.0 ‰ and -14.8 ± 7.3 ‰, while for $\Delta^{17}O(NO_3^-)$ the values are 28.8 ± 2.6 ‰ and 28.6 ± 3.2 ‰ respectively, suggesting that the inverse model reproduced the atmospheric observations quite well…"

[Figure]

**Figure 2.** Comparison between the observed and modeled annual averages of $\delta^{15}N(NO_3^-)$ and $\Delta^{17}O(NO_3^-)$ in the atmosphere at Summit, Greenland. The atmospheric observations are adapted from Jiang et al. (2022). The solid line in the box plot indicates the median value, while the dash line represents the average value.

*The authors may notice the new work by Shi et al. (2023), which is highly relevant to this manuscript and was just published after the submission of this manuscript. Please cite this work during the revision: Shi, G., Buffen, A. M., Hu, Y., Chai, J., Li, Y., Wang, D., & Hastings, M. G. (2023). Modeling the complete nitrogen and oxygen isotopic imprint of nitrate photolysis in snow. Geophysical Research Letters, 50, e2023GL103778*
**Response:** Thanks for this reminder. We have added this citation in the revised version.

---

## Author Comment (AC2)

We are grateful to the reviewer for the valuable suggestions and/or comments which improve the manuscript significantly. Below we list the detailed responses to the reviewer's suggestions and comments. The comments are listed in italics, followed by the response in normal font with changes highlighted in blue.

**Response to Referee #2**

*The paper by Jiang et al. presents for the first time an inverse model (based on the forward model from Erbland et al.) to reconstruct atmospheric nitrate load and its nitrogen AND oxygen isotopic signatures based on snow pack data. In particular, it includes the postdepositional loss/recycling of nitrate by photolysis and nitrate reformation and compares the results from two ice core end members    (Summit, Dome C) with atmospheric information. Overall, the results agree surprisingly well with atmospheric observations and for example support a clear stratospheric origin of the primary nitrate at Dome C. This all justifies the publication of this paper in ACP with minor revisions.*
**Response:** We are grateful to the reviewer for the time involved in reviewing the manuscript and for the encouraging comments on the merits of this work.

*Having said that, the paper is not always easy to follow and I am afraid that especially readers not familiar with the respective background of the mass balance and Rayleigh fractionation equations would need more guidance. I would therefore suggest to expand the Appendix A to give a more detailed derivation.*
**Response:** Thanks for this suggestion. We have added more detailed explanations on Eq(A2-4) in the Appendix A. The following text was added in the revised manuscript.
"    The direct photolysis of snow nitrate can be described by the Rayleigh equation. We define the first-order photolysis rate constant of $^{14}NO_3^-$ and $^{15}NO_3^-$ as $J$ and $J^*$ and their concentration in snow as $c$ and $c^*$ respectively. The chemical kinetic equations of $c$ and $c^*$ can be represented as follows:

$$\frac{dc}{dt} = -Jc \tag{A2}$$

$$\frac{dc^*}{dt} = -J^*c^* \tag{A3}$$

Integrating Eq(A2) and Eq(A3) yields Eq(A4) and Eq(A5):

$$c(t) = c(0)e^{-\int_0^t J dt} \tag{A4}$$

$$c^*(t) = c^*(0)e^{-\int_0^t J^* dt} \tag{A5}$$

Here $c(0)$ represents the initial concentration before photolysis. The evolution of the isotopic ratio $R$ which is defined as the ratio of $c$ and $c^*$ follows Eq(A6):

$$R(t) = \frac{c^*(t)}{c(t)} = \frac{c^*(0)}{c(0)}e^{-\int_0^t (J^*-J) dt} = R(0)e^{-\int_0^t (J^*-J) dt} \tag{A6}$$

Since the delta value $\delta^{15}N$ equals to $R_{spl}/R_{ref}-1$ where $R_{spl}$ and $R_{ref}$ refer to the isotope ratio of sample and standard respectively, Eq(A6) can be further expanded to:

$$ln\frac{1 + \delta(t)}{1 + \delta(0)} = ln\frac{R(t)}{R(0)} = -\int_0^t (J^* - J)dt$$

$$= -\int_0^t J\varepsilon_p dt = -\bar{\varepsilon}_p \int_0^t J dt = \bar{\varepsilon}_p ln(1 - f_p) \qquad (A7)$$

which is consistent with the form of the Rayleigh equation.

By applying the first-order Taylor expansion of $ln(1+\delta^{15}N(NO_3^-)) \approx \delta^{15}N(NO_3^-)$, we obtain the relationship between the $\delta^{15}N(NO_3^-)$ before and after photolysis:

$$\delta^{15}N(SN_r) \approx \delta^{15}N(SN') - \bar{\varepsilon}_p ln(1 - f_p) \qquad (A8)$$

The $\delta^{15}N$ of the emitted $NO_2$ can be calculate via the mass balance equation:

$$\delta^{15}N(SN') = (1 - f_p)\delta^{15}N(SN_r) + f_p\delta^{15}N(NO_2) \qquad (A9)$$

Combining Eq(A8) and Eq(A9) would yield:

$$\delta^{15}N(NO_2) \approx \delta^{15}N(SN') + \frac{\bar{\varepsilon}_p(1 - f_p)ln(1 - f_p)}{f_p} \qquad (A10)$$

Because part of the photoproduct would undergo cage effect to reform nitrate (Fig A1), the final state of snow $\delta^{15}N(NO_3^-)$ after photolysis can be calculated via isotopic mass balance equation:

$$\delta^{15}N(SN) = \frac{(1 - f_p)\delta^{15}N(SN_r) + f_c f_p\delta^{15}N(NO_2)}{1 - f_p + f_c f_p}$$

$$= \delta^{15}N(SN') - \frac{(1 - f_p)(1 - f_c)\bar{\varepsilon}_p \ln(1 - f_p)}{(1 - f_p) + f_c f_p} \qquad (A11)$$

which is equal to Eq (8)…"

*I also felt that the discussion of initial deposition and re-deposition of nitrate produced during photolysis needs somewhat more explanation in the beginning. In the end this process may easily explain, the observed deviations of the atmospheric d15N in observations and model results in certain months.*

**Response:** Thanks for this suggestion. We have added the following text in the introduction part:

"…These photoproducts subsequently reform nitrate (i.e., snow-sourced nitrate) and deposit locally or be exported away, leading to a recycling of nitrate at the air-snow interface (Erbland et al., 2013; Frey et al., 2009). The reformed nitrate would inherit $\Delta^{17}O$ signals under local oxidation conditions that is different from primary nitrate, and the re-deposition of atmospheric nitrate could also result in nitrogen isotopic fractionation depending on the different deposition mechanisms (Erbland et al., 2013; Jiang et al., 2022). Thus, post-depositional processing not only disturbs the link between nitrate in snow and its atmospheric precursors but also alters its isotopic signals…. But since these processes are initiated by sunlight, the post-depositional processing is muted in polar winter when sunlight is absent."

*Finally, a comparison with the results by Shi et al. in GRL (10.1029/2023GL103778), who also include oxygen isotopes in a forward model approach, is still missing in the discussion.*

**Response:** We noticed the publication of the Shi et al. paper after our manuscript was in discussion. Due to the similar topic, we have carefully examined the new work by Shi et al. (2023). From their paper and model source code (https://cstr.cn/18406.11. Cryos.tpdc.300476), we think the modeling approach in Shi et al. (2023) is basically the same as the TRANSITS model, except that Shi et al. (2023) extends the same procedures to simulate $\delta^{18}O(NO_3^-)$. We also note that a couple of years ago Joel Savarino provided the TRANSITS model code to Guitao Shi, the leading author of the Shi et al. (2023) study.

Since Shi et al. (2023) adopted a constant upper boundary condition for snowpack (i.e., constant deposited nitrate flux and isotopes), their model mainly focus on the pure photolytic effect on snow nitrate isotopes, which has been fully incorporated by the TRANSITS model. The only difference in their work is that the photolysis effect on snow $\delta^{18}O(NO_3^-)$ is considered, but we notice that Shi et al. (2023) model had to scale the theoretical fractionation factor ($^{18}\varepsilon_p$) to make the model results consistent with the observations. It remains unclear why the theoretical fractionation factor calculated using the ZPE shifted method (Frey et al., 2009) works well on $\delta^{15}N(NO_3^-)$ but not on $\delta^{18}O(NO_3^-)$. It appears to us that the uncertainty in the fractionation factors severally limits the extension of their method to other sites.

In summary, given the similarities in the modeling approach, logic, framework and others between Shi et al. (2023) and the TRANSITS model, it appears to us that it is not necessary to further compare our results with the Shi et al (2023) results, since we have already compared our model with the TRANSITS model.

In the revised manuscript, we have added the following citation of the Shi et al. (2023) with a brief discussion in the introduction after we introduced the TRANSITS model:

"…In addition, changes in the isotopic composition of nitrate ($\delta^{15}N$ and $\Delta^{17}O$) at each step of the post-depositional processing are also explicitly incorporated. Recently, Shi et al. (2023) extended or followed the TRANSITS model framework to include snowpack $\delta^{18}O(NO_3^-)$ simulation during the preservation of nitrate in snow. The latter was built upon the same chemical processes related to modeling $\Delta^{17}O(NO_3^-)$ changes during the post-depositional processing. However, the fractionation factor of $\delta^{18}O$ during snow nitrate photolysis ($^{18}\varepsilon_p$) had to be scaled to reproduce the observations. In this case it remains unclear why the theoretical fractionation factor calculated using the ZPE shifted method (Frey et al., 2009) works well on $\delta^{15}N(NO_3^-)$ but not on $\delta^{18}O(NO_3^-)$. Nevertheless, the uncertainties associated with $\delta^{18}O$ fractionations during snow nitrate photolysis and other processes (e.g., the cage effect, reformation of nitrate from $NO_2$, etc.) make this simulation less useful and reliable than for $\Delta^{17}O(NO_3^-)$, for which there are much less influencing factors and are easier to constrain…"

*Apart from this I made several comments and language corrections in the annotated pdf file attached.*

**Response:** Thanks for the detailed check. We have revised these typos accordingly in the main text. The response to each specified comment is listed below.

*Line 49: is it the ratio of O3/HOx in the atmosphere or also of their individual reaction rates with NOx. Please specify*

**Response:** Thanks for this question. It is the relative differences in the individual reaction rates that eventually determines the isotopes. However, since most of the related reactions are gas-phase reactions, basically it is mainly the relative concentrations of $O_3$ versus $HO_x$ that determines the rate differences (though the reaction rate constants are moderately dependent on temperature). In general, it can be approximated by the relative abundances of $O_3$ versus $HO_x$ (i.e., the ratio of $O_3/HO_x$) that determines $\Delta^{17}O(NO_3^-)$, as frequently used in literature.

*Line 223: weighted by what and averaged over what time scale?*

**Response:** In the inverse model, the algorithm to calculate the average $\varepsilon_p$ is described by the following equation:

$$\overline{\varepsilon_p} = \frac{\sum_0^{90} \varepsilon_p(sza)\Delta t(sza)}{\sum_0^{90} \Delta t(sza)} \tag{1}$$

In Eq(1), the $\varepsilon_p$ at different solar zenith angle (SZA) is first computed under the prescribed total column ozone. Then the average $\varepsilon_p$ for each week is computed by the weighted average of $\varepsilon_p$ over the duration of each SZA. Eq(1) can be regarded as the arithmetic mean of $\varepsilon_p$ over the entire week when the value is set to 0 if SZA is larger than 90 degrees. We changed the sentence as follows:

"…To simplify the calculation, in Eq. (8) $\overline{\varepsilon_p}$ in a certain week is calculated by the weighted average of nitrogen isotope fractionation constant over the durations of different solar zenith angles (0-90 degree)…"

*Lines 295-296: somewhere you need to specify what the depositon process of nitrate is. Is this gas phase adsorption or wet and dry deposition of particulate nitrate?*

**Response:** At the beginning of section 3, we add the following statement:

"…The deposited nitrate flux FD represents the state of nitrate that has just deposited onto the surface snow via dry deposition of gaseous nitrate or wet scavenge from the atmosphere and is close to the definition of the skin layer of snowpack…"

*Lines 347-350:This is confusing. Local midsummer at Dome C should be around week 0*

**Response:** In our model week 0 starts in winter (i.e, the first week of January in the northern hemisphere or the first week of July in the southern hemisphere). We emphasize this point at the beginning of this paragraph in the revised version:

"In Eq (20), $c_a$ represents the annual average snow nitrate concentration, $n$ represents the week number (1 to 52, here week 1 is defined as the first week in January for the

northern hemisphere sites or the first week in July for the southern hemisphere sites)
and the shape parameters ($a, b, \sigma$) were determined by the best fit of skin layer nitrate
concentrations (Appendix D)…"

*Line 366: is this assumption justified? How large is the measured seasonality in d15N
and D17O?*
**Response:** Thank you for this question. Unfortunately, by far there is no report on the
seasonality in $\delta^{15}N$ and $\Delta^{17}O$ of archived nitrate at Dome C due to the extremely low
snow accumulation rate which prevent high-resolution sampling to reveal the
snowpack seasonality, so technically we can't justify this assumption. However, we
speculate that the magnitude of seasonality (~ 20-30‰) in archived $\delta^{15}N(NO_3^-)$
should be much lower compared to the archived values (up to 334 ‰, Erbland et al.,
2013), and its impact should be small.

*Line 518: what exactly do you mean by summer snowpack? The profiles in Figure 5
cover a few years of snow deposition not just summer, so I assume you refer mainly to
the summer atmospheric NO3 concentration? Please explain*
**Response:** Sorry for the confusion, we meant to use "summer snowpack" to represent
the snowpack collected in summer. To avoid confusion, we delete the usage of
"summer snowpack" and just refer to as "snowpack"

*Line 568: do you mean: "are nor affected by" ??? the wording is confusing as
anything is irrelevant for a prescribed parameter as it is prescribed :-)*
**Response:** Thanks for pointing this out. It is a typo. We change the statement as
follow:
"…We note that the modeled isotopic compositions of snowpack and skin layer
nitrate are not affected by the prescribed nitrate concentration seasonality…"

*Line 690: looking at Fig. 8 I would say that the isotopic signature is largely
independent of fexp if fexp is larger than 0.1-0.2*
**Response:** We add the following statement for the sensitivity of $\delta^{15}N$:
"…For Dome C, the model results are sensitive to $f_{exp}$ when $f_c$ is small, and becomes
sensitive to $f_c$ when $f_{exp}$ is larger. In particular, the isotopic signature is largely
independent of $f_{exp}$ when $f_{exp}$ is larger than 0.1-0.2. In addition, the $\Delta^{17}O$ results
display a non-monotonic response to these two parameters…"

*Line 754: The language is incorrect here. fp denotes the total photolxzed NO3 while
fp x fc denotes the fraction photolyzed and caged as correctly displayed in A1*
**Response:** Thanks for pointing this out. We revised this sentence as follows:
"…As shown in Fig. A1, assuming a fraction ($f_p$) of initial snow nitrate was
photolyzed and a fraction ($f_c$) of these photolyzed nitrate has undergone the cage
effect, the mass balance equation for snow nitrate can be written as…"

*Line 763: This is not self-explanatory. What you refer to is that you made the approximation ln(1+d15N)= app. d15N. Spell it out*

**Response:** We change the sentence as follow:

"Here we apply the first-order Taylor expansion of $\ln(1+\delta^{15}N(NO_3^-)) \approx \delta^{15}N(NO_3^-)$."

*Line 766: there is a sign error in this equation . Plugging A2 and A3 into the first line of equation A4 (mass balance) would give (fc-1) not (1-fc) in the second line. In equ 8 it is correct*

**Response:** Thank you for this point. We have corrected this error in the revised version.